# Hausdorff Dimension, Heavy Tails, and Generalization in Neural Networks

**Umut Şimşekli[1,2], Ozan Sener[3], George Deligiannidis[2,4], Murat A. Erdogdu[5,6]**
LTCI, Télécom Paris, Institut Polytechnique de Paris[1], University of Oxford[2], Intel Labs[3]
The Alan Turing Institute[4], University of Toronto[5], Vector Institute[6]

## Abstract

Despite its success in a wide range of applications, characterizing the generalization properties of stochastic gradient descent (SGD) in non-convex deep learning problems is still an important challenge. While modeling the trajectories of SGD via stochastic differential equations (SDE) under heavy-tailed gradient noise has recently shed light over several peculiar characteristics of SGD, a rigorous treatment of the generalization properties of such SDEs in a learning theoretical framework is still missing. Aiming to bridge this gap, in this paper, we prove generalization bounds for SGD under the assumption that its trajectories can be well-approximated by a *Feller process*, which defines a rich class of Markov processes that include several recent SDE representations (both Brownian or heavy-tailed) as its special case. We show that the generalization error can be controlled by the *Hausdorff dimension* of the trajectories, which is intimately linked to the tail behavior of the driving process. Our results imply that heavier-tailed processes should achieve better generalization; hence, the tail-index of the process can be used as a notion of "capacity metric". We support our theory with experiments on deep neural networks illustrating that the proposed capacity metric accurately estimates the generalization error, and it does not necessarily grow with the number of parameters unlike the existing capacity metrics in the literature.

## 1 Introduction

Many important tasks in deep learning can be represented by the following optimization problem,

$$\min_{w \in \mathbb{R}^d} \Big\{ f(w) := \frac{1}{n} \sum_{i=1}^{n} f^{(i)}(w) \Big\}, \tag{1}$$

where $w \in \mathbb{R}^d$ denotes the network weights, $n$ denotes the number of training data points, $f$ denotes a non-convex cost function, and $f^{(i)}$ denotes the cost incurred by a single data point. Gradient-based optimization algorithms, perhaps Stochastic gradient descent (SGD) being the most popular one, have been the primary algorithmic choice for attacking such optimization problems. Given an initial point $w_0$, the SGD algorithm is based on the following recursion,

$$w_{k+1} = w_k - \eta \nabla \tilde{f}_k(w_k) \quad \text{with} \quad \nabla \tilde{f}_k(w) := \frac{1}{B} \sum_{i \in \tilde{B}_k} \nabla f^{(i)}(w), \tag{2}$$

where $\eta$ is the step-size, and $\nabla \tilde{f}_k$ is the unbiased stochastic gradient with batch size $B = |\tilde{B}_k|$ for a random subset $\tilde{B}_k$ of $\{1, \ldots, n\}$ for all $k \in \mathbb{N}$, $|\cdot|$ denoting cardinality.

In contrast to convex optimization setting where the behavior of SGD is fairly well-understood (see e.g. [DDB19, SSBD14]), the generalization properties of SGD in non-convex deep learning problems is an active area of research [PBL19, AZL19, AZLL19]. In the last decade, there has been considerable progress around this topic, where several generalization bounds have been proven in

different mathematical setups [NTS15, MWZZ17, Lon17, DR17, KL17, RRT17, ZLZ19, AZLL19, NHD$^+$19]. While these bounds are useful at capturing the generalization behavior of SGD in certain cases, they typically grow with dimension $d$, which contradicts empirical observations [NBMS17].

An important initial step towards developing a concrete generalization theory for the SGD algorithm in deep learning problems, is to characterize the statistical properties of the weights $\{w_k\}_{k\in\mathbb{N}}$, as they might provide guidance for identifying the constituents that determine the performance of SGD. A popular approach for analyzing the dynamics of SGD, mainly borrowed from statistical physics, is based on viewing it as a discretization of a *continuous-time* stochastic process that can be described by a stochastic differential equation (SDE). For instance, if we assume that the gradient noise, i.e., $\nabla\tilde{f}_k(w) - \nabla f(w)$ can be well-approximated with a Gaussian random vector, we can represent (2) as the Euler-Maruyama discretization of the following SDE,

$$\mathrm{dW}_t = -\nabla f(\mathrm{W}_t)\mathrm{d}t + \Sigma(\mathrm{W}_t)\mathrm{dB}_t, \tag{3}$$

where $\mathrm{B}_t$ denotes the standard Brownian motion in $\mathbb{R}^d$, and $\Sigma : \mathbb{R}^d \mapsto \mathbb{R}^{d\times d}$ is called the diffusion coefficient. This approach has been adopted by several studies [MHB16, JKA$^+$17, HLLL17, CS18, ZWY$^+$19]. In particular, based on the 'flat minima' argument (cf. [HS97]), Jastrzebski et al. [JKA$^+$17] illustrated that the performance of SGD on unseen data correlates well with the ratio $\eta/B$.

More recently, Gaussian approximation for the gradient noise has been taken under investigation. While Gaussian noise can accurately characterize the behavior of SGD for very large batch sizes [PSGN19], Simsekli et. al. [SSG19] empirically demonstrated that the gradient noise in fully connected and convolutional neural networks can exhibit *heavy-tailed* behavior in practical settings. This characteristic was also observed in recurrent neural networks [ZKV$^+$19]. Favaro et al. [FFP20] illustrated that *the iterates themselves* can exhibit heavy-tails and investigated the corresponding asymptotic behavior in the infinite-width limit. Similarly, Martin and Mahoney [MM19] observed that the eigenspectra of the weight matrices in individual layers of a neural network can exhibit heavy-tails; hence, they proposed a layer-wise heavy-tailed model for the SGD iterates. By invoking results from heavy-tailed random matrix theory, they proposed a capacity metric based on a quantification of the heavy-tails, which correlated well with the performance of the network on unseen data. Further, they empirically demonstrated that this capacity metric does not necessarily grow with dimension $d$.

Based on the argument that the observed heavy-tailed behavior of SGD[1] in practice cannot be accurately represented by an SDE driven by a Brownian motion, Simsekli et al. [SSG19] proposed modeling SGD with an SDE driven by a heavy-tailed process, so-called the $\alpha$-stable Lévy motion [Sat99]. By using this framework and invoking metastability results proven in statistical physics [IP06, Pav07], SGD is shown to spend more time around 'wider minima', and the time spent around those minima is linked to the tail properties of the driving process [SSG19, NŞGR19].

Even though the SDE representations of SGD have provided many insights on several distinguishing characteristics of this algorithm in deep learning problems, a rigorous treatment of their generalization properties in a statistical learning theoretical framework is still missing. In this paper, we aim to take a first step in this direction and prove novel generalization bounds in the case where the trajectories of the optimization algorithm (including but not limited to SGD) can be well-approximated by a *Feller process* [Sch16], which form a broad class of Markov processes that includes many important stochastic processes as a special case. More precisely, as a proxy to SGD, we consider the Feller process that is expressed by the following SDE:

$$\mathrm{dW}_t = -\nabla f(\mathrm{W}_t)\mathrm{d}t + \Sigma_1(\mathrm{W}_t)\mathrm{dB}_t + \Sigma_2(\mathrm{W}_t)\mathrm{dL}_t^{\boldsymbol{\alpha}(\mathrm{W}_t)}, \tag{4}$$

where $\Sigma_1, \Sigma_2$ are $d \times d$ matrix-valued functions, and $\mathrm{L}_t^{\boldsymbol{\alpha}(\cdot)}$ denotes the *state-dependent* $\boldsymbol{\alpha}$-stable Lévy motion, which will be defined in detail in Section 2. Informally, $\mathrm{L}_t^{\boldsymbol{\alpha}(\cdot)}$ can be seen as a heavy-tailed generalization of the Brownian motion, where $\boldsymbol{\alpha} : \mathbb{R}^d \mapsto (0,2]^d$ denotes its state-dependent *tail-indices*. In the case $\alpha_i(w) = 2$ for all $i$ and $w$, $\mathrm{L}_t^{\boldsymbol{\alpha}(\cdot)}$ reduces to $\sqrt{2}\mathrm{B}_t$ whereas if $\alpha_i$ gets smaller than 2, the process becomes heavier-tailed in the $i$-th component, whose tails asymptotically obey a power-law decay with exponent $\alpha_i$. The SDEs in [MHB16, JKA$^+$17, HLLL17, CS18, ZWY$^+$19]

all appear as a special case of (4) with $\Sigma_2 = 0$, and the SDE proposed in [SSG19] corresponds to the isotropic setting: $\Sigma_2(w)$ is diagonal and $\alpha_i(w) = \alpha \in (0,2]$ for all $i, w$. In (4), we allow each coordinate of $\mathrm{L}_t^{\boldsymbol{\alpha}(\cdot)}$ to have a different tail-index which can also depend on the state $\mathrm{W}_t$. We believe that $\mathrm{L}_t^{\boldsymbol{\alpha}(\cdot)}$ provides a more realistic model based on the empirical results of [ŞGN$^+$19], suggesting that the tail index can have different values at each coordinate and evolve over time.

At the core of our approach lies the fact that the sample paths of Markov processes often exhibit a *fractal-like* structure [Xia03], and the generalization error over the sample paths is intimately related to the 'roughness' of the random fractal generated by the driving Markov process, as measured by a notion called the Hausdorff dimension. Our main contributions are as follows.

($i$)    We introduce a novel notion of complexity for the trajectories of a stochastic learning algorithm, which we coin as 'uniform Hausdorff dimension'. Building on [Sch98], we show that the sample paths of Feller processes admit a uniform Hausdorff dimension, which is closely related to the tail properties of the process.

($ii$)    By using tools from geometric measure theory, we prove that the generalization error can be controlled by the Hausdorff dimension of the process, which can be significantly smaller than the standard Euclidean dimension. In this sense, the Hausdorff dimension acts as an 'intrinsic dimension' of the problem, mimicking the role of Vapnik-Chervonenkis (VC) dimension in classical generalization bounds.

These two contributions collectively show that heavier-tailed processes achieve smaller generalization error, implying that the heavy-tails of SGD incur an implicit regularization. Our results also provide a theoretical justification to the observations reported in [MM19] and [SSG19]. Besides, a remarkable feature of the Hausdorff dimension is that it solely depends on the tail behavior of the process; hence, contrary to existing capacity metrics, it does not necessarily grow with the number of parameters $d$. Furthermore, we provide an efficient approach to estimate the Hausdorff dimension by making use of existing tail index estimators, and empirically demonstrate the validity of our theory on various neural networks. Experiments on both synthetic and real data verify that our bounds do not grow with the problem dimension, providing an accurate characterization of the generalization performance.

## 2    Technical Background

**Stable distributions.**    Stable distributions appear as the limiting distribution in the generalized central limit theorem [Lév37] and can be seen as a generalization of the Gaussian distribution. In this paper, we will be interested in *symmetric* $\alpha$-stable distributions, denoted by $\mathcal{S}\alpha\mathcal{S}$. In the one-dimensional case, a random variable $X$ is $\mathcal{S}\alpha\mathcal{S}(\sigma)$ distributed, if its characteristic function (chf.) has the following form: $\mathbb{E}[\exp(i\omega X)] = \exp(-|\sigma\omega|^\alpha)$, where $\alpha \in (0,2]$ is called the *tail-index* and $\sigma \in \mathbb{R}_+$ is called the *scale* parameter. When $\alpha = 2$, $\mathcal{S}\alpha\mathcal{S}(\sigma) = \mathcal{N}(0, 2\sigma^2)$, where $\mathcal{N}$ denotes the Gaussian distribution in $\mathbb{R}$. As soon as $\alpha < 2$, the distribution becomes heavy-tailed and $\mathbb{E}[|X|^q]$ becomes finite if and only if $q < \alpha$, indicating that the variance of $\mathcal{S}\alpha\mathcal{S}$ is finite only when $\alpha = 2$.

There are multiple ways to extend $\mathcal{S}\alpha\mathcal{S}$ to the multivariate case. In our experiments, we will be mainly interested in the *elliptically-contoured* $\alpha$-stable distribution [ST94], whose chf. is given by $\mathbb{E}[\exp(i\langle\omega, X\rangle)] = \exp(-\|\omega\|^\alpha)$ for $X, \omega \in \mathbb{R}^d$, where $\langle\cdot,\cdot\rangle$ denotes the Euclidean inner product. Another common choice is the multivariate $\boldsymbol{\alpha}$-stable distribution *with independent components* for a vector $\boldsymbol{\alpha} \in \mathbb{R}^d$, whose chf. is given by $\mathbb{E}[\exp(i\langle\omega, X\rangle)] = \exp(-\sum_{i=1}^d |\omega_i|^{\alpha_i})$. Essentially, the $i$-th component of $X$ is distributed with $\mathcal{S}\alpha\mathcal{S}$ with parameters $\alpha_i$ and $\sigma_i = 1$. Both of these multivariate distributions reduce to a multivariate Gaussian when their tail indices are 2.

**Lévy and Feller processes.**    We begin by defining a general Lévy process (also called Lévy motion), which includes Brownian motion $\mathrm{B}_t$ and the $\alpha$-stable motion $\mathrm{L}_t^\alpha$ as special cases[2]. A Lévy process $\{\mathrm{L}_t\}_{t\geq 0}$ in $\mathbb{R}^d$ with the initial point $\mathrm{L}_0 = 0$, is defined by the following properties:

(i)    For $N \in \mathbb{N}$ and $t_0 < t_1 < \cdots < t_N$, the increments $(\mathrm{L}_{t_i} - \mathrm{L}_{t_{i-1}})$ are independent for all $i$.

(ii)    For any $t > s > 0$, $(\mathrm{L}_t - \mathrm{L}_s)$ and $\mathrm{L}_{t-s}$ have the same distribution.

(iii)    $\mathrm{L}_t$ is continuous in probability, i.e., for all $\delta > 0$ and $s \geq 0$, $\mathbb{P}(|\mathrm{L}_t - \mathrm{L}_s| > \delta) \to 0$ as $t \to s$.

By the Lévy-Khintchine formula [Sat99], the chf. of a Lévy process is given by $\mathbb{E}[\exp(i\langle\xi, \mathrm{L}_t\rangle)] = \exp(-t\psi(\xi))$, where $\psi : \mathbb{R}^d \mapsto \mathbb{C}$ is called the characteristic (or Lévy) exponent, given as:

$$\psi(\xi) = i\langle b, \xi\rangle + \frac{1}{2}\langle\xi, \Sigma\xi\rangle + \int_{\mathbb{R}^d}\left[1 - e^{i\langle x,\xi\rangle} + \frac{i\langle x,\xi\rangle}{1 + \|x\|^2}\right]\nu(\mathrm{d}x), \quad \forall\xi \in \mathbb{R}^d. \tag{5}$$

Here, $b \in \mathbb{R}^d$ denotes a constant drift, $\Sigma \in \mathbb{R}^{d\times d}$ is a positive semi-definite matrix, and $\nu$ is called the Lévy measure, which is a Borel measure on $\mathbb{R}^d \setminus \{0\}$ satisfying $\int_{\mathbb{R}^d}\|x\|^2/(1 + \|x\|^2)\nu(\mathrm{d}x) < \infty$. The choice of $(b, \Sigma, \nu)$ determines the law of $\mathrm{L}_{t-s}$; hence, it fully characterizes the process $\mathrm{L}_t$ by the Properties (i) and (ii) above. For instance, from (5), we can easily verify that under the choice $b = 0$, $\Sigma = \frac{1}{2}\mathrm{I}_d$, and $\nu(\xi) = 0$, with $\mathrm{I}_d$ denoting the $d \times d$ identity matrix, the function $\exp(-\psi(\xi))$ becomes the chf. of a standard Gaussian in $\mathbb{R}^d$; hence, $\mathrm{L}_t$ reduces to $\mathrm{B}_t$. On the other hand, if we choose $b = 0$, $\Sigma = 0$, and $\nu(\mathrm{d}x) = \frac{\mathrm{d}r}{r^{1+\alpha}}\lambda(\mathrm{d}y)$, for all $x = ry, (r, y) \in \mathbb{R}_+ \times \mathbb{S}^{d-1}$ where $\mathbb{S}^{d-1}$ denotes unit sphere in $\mathbb{R}^d$ and $\lambda$ is an arbitrary Borel measure on $\mathbb{S}^{d-1}$, we obtain the chf. of a generic multivariate $\alpha$-stable distribution, hence $\mathrm{L}_t$ reduces to $\mathrm{L}_t^\alpha$. Depending on $\lambda$, $\exp(-\psi(\xi))$ becomes the chf. of an elliptically contoured $\alpha$-stable distribution or an $\alpha$-stable distribution with independent components [Xia03].

Feller processes (also called *Lévy-type processes* [BSW13]) are a general family of Markov processes, which further extend the scope of Lévy processes. In this study, we consider a class of Feller processes [Cou65], which locally behave like Lévy processes and they additionally allow for *state-dependent* drifts $b(w)$, diffusion matrices $\Sigma(w)$, and Lévy measures $\nu(w, \mathrm{d}y)$ for $w \in \mathbb{R}^d$. For a fixed state $w$, a Feller process $\{\mathrm{W}_t\}_{t\geq 0}$ is defined through the chf. of the random variable $\mathrm{W}_t - w$, given as $\psi_t(w, \xi) = \mathbb{E}\left[\exp(-i\langle\xi, \bar{\mathrm{W}}_t - w\rangle)\right]$. A crucial characteristic of a Feller process related to its chf. is its *symbol* $\Psi$, defined as,

$$\Psi(w, \xi) = i\langle b(w), \xi\rangle + \frac{1}{2}\langle\xi, \Sigma(w)\xi\rangle + \int_{\mathbb{R}^d}\left[1 - e^{i\langle x,\xi\rangle} + \frac{i\langle w,\xi\rangle}{1 + \|x\|^2}\right]\nu(w, \mathrm{d}x), \tag{6}$$

for $w, \xi \in \mathbb{R}^d$ [Sch98, Xia03, Jac02]. Here, for each $w \in \mathbb{R}^d$, $\Sigma(w) \in \mathbb{R}^{d\times d}$ is symmetric positive semi-definite, and for all $w$, $\nu(w, \mathrm{d}x)$ is a Lévy measure.

Under mild conditions, one can verify that the SDE (4) we use as a proxy for the SGD algorithm indeed corresponds to a Feller process with $b(w) = -\nabla f(w)$, $\Sigma(w) = 2\Sigma_1(w)$, and an appropriate choice of $\nu$ (see [HDS18]). We also note that many other popular stochastic optimization algorithms can be accurately represented by a Feller process, which we describe in the supplementary document. Hence, our results can be useful in a broader context.

**Decomposable Feller processes.** In this paper, we will focus on decomposable Feller processes introduced in [Sch98], which will be useful in both our theory and experiments. Let $\mathrm{W}_t$ be a Feller process with symbol $\Psi$. We call the process $\mathrm{W}_t$ *'decomposable at $w_0$'*, if there exists a point $w_0 \in \mathbb{R}^d$, such that $\Psi(w, \xi) = \psi(\xi) + \tilde{\Psi}(w, \xi)$, where $\psi(\xi) := \Psi(w_0, \xi)$ is called the *sub-symbol* and $\tilde{\Psi}(w, \xi) := \Psi(w, \xi) - \Psi(w_0, \xi)$ is the remainder term. Here, $\tilde{\Psi}$ is assumed to satisfy certain smoothness and boundedness assumptions, which are provided in the supplementary document. Essentially, the technical regularity conditions on $\tilde{\Psi}$ impose a structure on the triplet $(b, \Sigma, \nu)$ around $w_0$ which ensures that, around that point, $\mathrm{W}_t$ behaves like a Lévy process whose characteristic exponent is given by the sub-symbol $\psi$.

**The Hausdorff Dimension.** Due to their recursive nature, Markov processes often generate 'random fractals' [Xia03] and understanding the structure of such fractals has been a major challenge in modern probability theory [BP17, Kho09, KX17, Yan18, LG19, LY19]. In this paper, we are interested in identifying the complexity of the fractals generated by a Feller process that approximates SGD.

The *intrinsic complexity* of a fractal is typically characterized by a notion called the Hausdorff dimension [Fal04], which extends the usual notion of dimension (e.g., a line segment is one-dimensional, a plane is two-dimensional) to fractional orders. Informally, this notion measures the 'roughness' of an object (i.e., a set) and in the context of Lévy processes, they are deeply connected to the tail properties of the corresponding Lévy measure. [Sch98, Xia03, Yan18].

Before defining the Hausdorff dimension, we need to introduce the Hausdorff measure. Let $G \subset \mathbb{R}^d$ and $\delta > 0$, and consider all the $\delta$-coverings $\{A_i\}_i$ of $G$, i.e., each $A_i$ denotes a set with diameter less than $\delta$ satisfying $G \subset \cup_i A_i$. For any $s \in (0, \infty)$, we then denote: $\mathcal{H}_\delta^s(G) := \inf\sum_{i=1}^\infty \mathrm{diam}(A_i)^s$, where the infimum is taken over all the $\delta$-coverings. The $s$-dimensional Hausdorff measure of $G$

is defined as the monotonic limit: $\mathcal{H}^s(G) := \lim_{\delta \to 0} \mathcal{H}^s_\delta(G)$. It can be shown that $\mathcal{H}^s$ is an outer measure; hence, it can be extended to a complete measure by the Carathéodory extension theorem [Mat99]. When $s$ is an integer, $\mathcal{H}^s$ is equal to the $s$-dimensional Lebesgue measure up to a constant factor; thus, it strictly generalizes the notion 'volume' to the fractional orders. We now proceed with the definition of the Hausdorff dimension.

**Definition 1.** *The Hausdorff dimension of $G \subset \mathbb{R}^d$ is defined as follows.*

$$\dim_{\mathrm{H}} G := \sup\{s > 0 : \mathcal{H}^s(G) > 0\} = \inf\{s > 0 : \mathcal{H}^s(G) < \infty\}. \tag{7}$$

One can show that if $\dim_{\mathrm{H}} G = s$, then $\mathcal{H}^r(G) = 0$ for all $r > s$ and $\mathcal{H}^r(G) = \infty$ for all $r < s$ [EMG90]. In this sense, the Hausdorff dimension of $G$ is the moment order $s$ when $\mathcal{H}^s(G)$ drops from $\infty$ to 0, and we always have $0 \le \dim_{\mathrm{H}} G \le d$ [Fal04]. Apart from the trivial cases such as $\dim_{\mathrm{H}} \mathbb{R}^d = d$, a canonical example is the well-known Cantor set, whose Hausdorff dimension is $(\log 2 / \log 3) \in (0, 1)$. Besides, the Hausdorff dimension of Riemannian manifolds correspond to their intrinsic dimension, e.g., $\dim_{\mathrm{H}} \mathbb{S}^{d-1} = d - 1$.

We note that, starting with the seminal work of Assouad [Ass83], tools from fractal geometry have been considered in learning theory [SHTY13, MSS19, DSD19] in different contexts. In this paper, we consider the Hausdorff dimension of the sample paths of Markov processes in a learning theoretical framework, which, to the best of our knowledge, has not yet been investigated in the literature.

## 3  Uniform Hausdorff Dimension and Generalization

**Mathematical setup.**  Let $\mathcal{Z} = \mathcal{X} \times \mathcal{Y}$ denote the space of data points, with $\mathcal{X}$ being the space of features and $\mathcal{Y}$ the space of the labels. We consider an unknown data distribution over $\mathcal{Z}$, denoted by $\mu_z$. We assume that we have access to a training set with $n$ elements, denoted as $S = \{z_1, \dots, z_n\}$, where each element of $S$ is independently and identically distributed (i.i.d.) from $\mu_z$. We will denote $S \sim \mu_z^{\otimes n}$, where $\mu_z^{\otimes n}$ is the $n$-times product measure of $\mu_z$.

To assess the quality of an estimated parameter, we consider a loss function $\ell : \mathbb{R}^d \times \mathcal{Z} \mapsto \mathbb{R}_+$, such that $\ell(w, z)$ measures the loss induced by a single data point $z$ for the particular choice of parameter $w \in \mathbb{R}^d$. We accordingly denote the population risk with $\mathcal{R}(w) := \mathbb{E}_z[\ell(w, z)]$ and the empirical risk with $\hat{\mathcal{R}}(w, S) := \frac{1}{n} \sum_{i=1}^n \ell(w, z_i)$. We note that we allow the cost function $f$ in (1) and the loss $\ell$ to be different from each other, where $f$ should be seen as a surrogate loss function. In particular, we will have different sets of assumptions on $f$ and $\ell$. However, as $f$ and $\ell$ are different from each other, the discrepancy between the risks of their respective minimizers would have an impact on generalization. We leave the analysis of such discrepancy as a future work.

An *iterative training algorithm* $\mathcal{A}$ (for example SGD) is a function of two variables $S$ and $U$, where $S$ denoting the dataset and $U$ encapsulating all the *algorithmic randomness* (e.g. batch indices to be used in training). The algorithm $\mathcal{A}(S, U)$ returns the entire evolution of the parameters in the time frame $[0, T]$, where $[\mathcal{A}(S, U)]_t = w_t$ being the parameter value returned by $\mathcal{A}$ at time $t$ (e.g. parameters trained by SGD at time $t$). More precisely, given a training set $S$ and a random variable $U$, the algorithm will output a random process $\{w_t\}_{t \in [0,T]}$ indexed by time, which is the *trajectory of iterates*. To formalize this definition, let us denote the class of bounded Borel functions defined from $[0, T]$ to $\mathbb{R}^d$ with $\mathcal{B}([0, T], \mathbb{R}^d)$, and define $\mathcal{A} : \mathcal{Z}^n \times \Omega \mapsto \mathcal{B}([0, T], \mathbb{R}^d)$, where $\Omega$ denotes the domain of $U$. We will denote the law of $U$ by $\mu_u$, and without loss of generality we let $T = 1$.

In the remainder of the paper, we will consider the case where the algorithm $\mathcal{A}$ is chosen to be the trajectories produced by a Feller process $\mathrm{W}^{(S)}$ (e.g. the proxy for SGD (4)), whose symbol depends on the training set $S$. More precisely, given $S \in \mathcal{Z}^n$, the output of the training algorithm $\mathcal{A}(S, \cdot)$ will be the random mapping $t \mapsto \mathrm{W}_t^{(S)}$, where the symbol of $\mathrm{W}^{(S)}$ is determined by the drift $b_S(w)$, diffusion matrix $\Sigma_S(w)$, and the Lévy measure $\nu_S(w, \cdot)$ (see (6) for definitions), which all depend on $S$. In this context, the random variable $U$ represents the randomness that is incurred by the Feller process. In particular, for the SDE proxy (4), $U$ accounts for the randomness due to $\mathrm{B}_t$ and $\mathrm{L}_t^{\alpha(\cdot)}$.

As our framework requires $\mathcal{A}$ to produce continuous-time trajectories to represent the discrete-time recursion of SGD (2), we can consider the *linearly interpolated* continuous-time process; an approach which is commonly used in SDE analysis [Dal17, RRT17, EMS18, NŞGR19, EH20]. For a given $t \in [k\eta, (k+1)\eta)$, we can define the process $\tilde{\mathrm{W}}_t$ as the linear interpolation of $w_k$ and $w_{k+1}$ (see

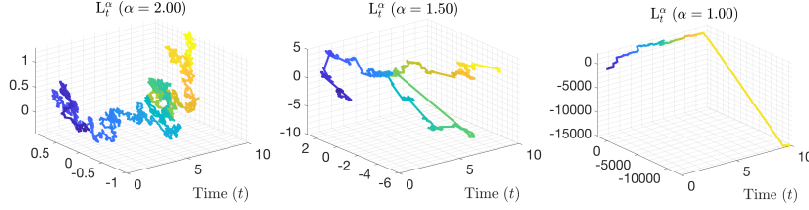

Figure 1: Trajectories of $L_t^\alpha$ for $\alpha = 2.0$, $1.5$, and $1.0$. The colors indicate the evolution in time. We observe that the trajectories become 'simpler' as $\dim_H L^\alpha[0, T] = \alpha$ gets smaller.

(2)), such that $w_k = \tilde{W}_{k\eta}$ for all $k$. On the other hand, the random variable $U$ here represents the randomness incurred by the choice of the random minibatches $\tilde{B}_k$ over iterations (2).

Throughout this paper, we will assume that $S$ and $U$ are independent from each other. In the case of (4), this will entail that the randomness in $B_t$ and $L_t^\alpha$ does not depend on $S$, or in the case of the SGD recursion, it will require the random sets $\tilde{B}_k \subset \{1, \dots, n\}$ to be drawn independently from $S$[3]. Under this assumption, $U$ does not play a crucial role in our analysis; hence, to ease the notation, we will occasionally omit the dependence on $U$ and simply write $\mathcal{A}(S) := \mathcal{A}(S, U)$. We will further use the notation $[\mathcal{A}(S)]_t := [\mathcal{A}(S, U)]_t$ to refer to $w_t$. Without loss of generality, we will assume that the training algorithm is always initialized with zeros, i.e., $[\mathcal{A}(S)]_0 = \mathbf{0} \in \mathbb{R}^d$, for all $S \in \mathcal{Z}^n$. Finally, we define the collection of the parameters given in a trajectory, as the image of $\mathcal{A}(S)$, i.e., $\mathcal{W}_S := \{w \in \mathbb{R}^d : \exists t \in [0,1], w = [\mathcal{A}(S)]_t\}$ and the collection of all possible parameters as the union $\mathcal{W} := \bigcup_{S \in \mathcal{Z}^n} \mathcal{W}_S$. Note that $\mathcal{W}$ is still random due to its dependence on $U$.

**Uniform Hausdorff dimension and Feller processes.** In this part, we introduce the 'uniform Hausdorff dimension' property for a training algorithm $\mathcal{A}$, which is a notion of complexity based on the Hausdorff dimension of the trajectories generated by $\mathcal{A}$. By translating [Sch98] into our context, we will then show that decomposable Feller processes possess this property.

**Definition 2.** *An algorithm $\mathcal{A}$ has uniform Hausdorff dimension $d_H$ if for any training set $S \in \mathcal{Z}^n$*

$$\dim_H \mathcal{W}_S = \dim_H\{w \in \mathbb{R}^d : \exists t \in [0,1], w = [\mathcal{A}(S, U)]_t\} \leq d_H, \quad \mu_u\text{-almost surely.} \quad (8)$$

Since $\mathcal{W}_S \subset \mathcal{W} \subset \mathbb{R}^d$, by the definition of Hausdorff dimension, any algorithm $\mathcal{A}$ possesses the uniform Hausdorff dimension property trivially with $d_H = d$. However, as we will illustrate in the sequel, $d_H$ can be much smaller than $d$, which is of our interest in this study.

**Proposition 1.** *Let $\{W^{(S)}\}_{S \in \mathcal{Z}^n}$ be a family of Feller processes. Assume that for each $S$, $W^{(S)}$ is decomposable at a point $w_S$ with sub-symbol $\psi_S$. Consider the algorithm $\mathcal{A}$ that returns $[\mathcal{A}(S)]_t = W_t^{(S)}$ for a given $S \in \mathcal{Z}^n$ and for every $t \in [0,1]$. Then, we have*

$$\dim_H \mathcal{W}_S \leq \beta_S, \quad \text{where} \quad \beta_S := \inf\left\{\lambda \geq 0 : \lim_{\|\xi\| \to \infty} \frac{|\psi_S(\xi)|}{\|\xi\|^\lambda} = 0\right\}, \quad (9)$$

*$\mu_u$-almost surely. Furthermore, $\mathcal{A}$ has uniform Hausdorff dimension with $d_H = \sup_{S \in \mathcal{Z}^n} \beta_S$.*

We provide all the proofs in the supplementary document. Informally, this result can be interpreted as follows. Thanks to the decomposability property, for each $S$, the process $W^{(S)}$ behaves like a Lévy motion around $w_S$, and the characteristic exponent is given by the sub-symbol $\psi_S$. Because of this locally regular behavior, the Hausdorff dimension of the image of $W^{(S)}$ can be bounded by $\beta_S$, which only depends on *tail behavior* of the Lévy process whose exponent is the sub-symbol $\psi_S$.

**Example 1.** *In order to illustrate Proposition 1, let us consider a simple example, where $W_t^{(S)}$ is taken as the $d$-dimensional $\alpha$-stable process with $d \geq 2$, which is independent of the data sample $S$. More precisely, $W_t^{(S)}$ is the solution to the SDE given by $dW_t^{(S)} = dL_t^\alpha$ for some $\alpha \in (0, 2]$, where $L_1^\alpha$ is an elliptically-contoured $\alpha$-stable random vector. As $W_t^{(S)}$ is already a Lévy process, it trivially satisfies the assumptions of Proposition 1 with $\beta_S = \alpha$ for all $S$ [BG60], hence $\dim_H \mathcal{W}_S \leq \alpha$, $\mu_u$-almost surely (in fact, one can show that $\dim_H \mathcal{W}_S = \alpha$, see [BG60], Theorem 4.2). Hence, the 'algorithm' $[\mathcal{A}(S)]_t = W_t^{(S)}$ has uniform Hausdorff dimension $d_H = \alpha$. This shows that as the process becomes heavier-tailed (i.e., $\alpha$ decreases), the Hausdorff dimension $d_H$ gets smaller. This behavior is illustrated in Figure 1.*

The term $\beta_S$ is often termed as the upper Blumenthal-Getoor (BG) index of the Lévy process with an exponent $\psi_S$ [BG60], and it is directly related to the *tail-behavior* of the corresponding Lévy measure. In general, the value of $\beta_S$ decreases as the process gets heavier-tailed, which implies that the heavier-tailed processes have smaller Hausdorff dimension; thus, they have smaller complexity.

**Generalization bounds via Hausdorff dimension.** This part provides the main contribution of this paper, where we show that the generalization error of a training algorithm can be controlled by the Hausdorff dimension of its trajectories. Even though our interest is still in the case where $\mathcal{A}$ is chosen as a Feller process, the results in this section apply to more general algorithms. To this end, we will be mainly interested in bounding the following object:

$$\sup_{t\in[0,1]} |\hat{\mathcal{R}}([\mathcal{A}(S)]_t, S) - \mathcal{R}([\mathcal{A}(S)]_t)| = \sup_{w\in\mathcal{W}_S} |\hat{\mathcal{R}}(w,S) - \mathcal{R}(w)|, \tag{10}$$

with high probability over the choice of $S$ and $U$. Note that this is an algorithm dependent definition of generalization that is widely used in the literature (see [BE02] for a detailed discussion).

To derive our first result, we will require the following assumptions.

**H 1.** *$\ell$ is bounded by $B$ and $L$-Lipschitz continuous in $w$.*

**H 2.** *The diameter of $\mathcal{W}$ is finite $\mu_u$-almost surely. $S$ and $U$ are independent.*

**H 3.** *$\mathcal{A}$ has uniform Hausdorff dimension $d_{\mathrm{H}}$.*

**H 4.** *For $\mu_u$-almost every $\mathcal{W}$, there exists a Borel measure $\mu$ on $\mathbb{R}^d$ and positive numbers $a, b, r_0$ and $s$ such that $0 < \mu(\mathcal{W}) \le \mu(\mathbb{R}^d) < \infty$ and $0 < ar^s \le \mu(B_d(x,r)) \le br^s < \infty$ for $x \in \mathcal{W}, 0 < r \le r_0$.*

Boundedness of the loss can be relaxed at the expense of using sub-Gaussian concentration bounds and introducing more complexity into the expressions [MBM16]. More precisely, **H**1 can be replaced with the assumption $\exists K > 0$, such that $\forall p$, $\mathbb{E}[\ell(w,z)^p]^{1/p} \le K\sqrt{p}$, and by using sub-Gaussian concentration our bounds will still hold with $K$ in place of $B$. On the other hand, since we have a finite time-horizon and we fix the initial point of the processes to $\mathbf{0}$, by using [XZ20] Lemma 7.1, we can show that the finite diameter condition on $\mathcal{W}$ holds almost surely, if standard regularity assumptions hold uniformly on the coefficients of $\mathrm{W}^{(S)}$ (i.e., $b$, $\Sigma$, and $\nu$ in (6)) for all $S \in \mathcal{Z}^n$, and a countability condition on $\mathcal{Z}$. Finally **H**4 is a common condition in fractal geometry, and ensures that the set $\mathcal{W}$ is regular enough, so that we can relate its Hausdorff dimension to its covering numbers [Mat99][4]. Under these conditions and an additional countability condition on $\mathcal{Z}$ (see [BE02] for similar assumptions), we present our first main result as follows.

**Theorem 1.** *Assume that **H**1 to 4 hold, and $\mathcal{Z}$ is countable. Then, for a sufficiently large $n$, we have*

$$\sup_{w\in\mathcal{W}_S} |\hat{\mathcal{R}}(w,S) - \mathcal{R}(w)| \le B\sqrt{\frac{2d_{\mathrm{H}}\log(nL^2)}{n} + \frac{\log(1/\gamma)}{n}}, \tag{11}$$

*with probability at least $1 - \gamma$ over $S \sim \mu_z^{\otimes n}$ and $U \sim \mu_u$.*

This theorem shows that the generalization error can be controlled by the uniform Hausdorff dimension of the algorithm $\mathcal{A}$, along with the constants inherited from the regularity conditions. A noteworthy property of this result is that it does not have a direct dependency on the number of parameters $d$; on the contrary, we observe that $d_{\mathrm{H}}$ plays the role that $d$ plays in standard bounds [AB09], implying that $d_{\mathrm{H}}$ acts as the *intrinsic dimension* and mimics the role of the VC dimension in binary classification [SSBD14]. Furthermore, in combination with Proposition 1 that indicates $d_{\mathrm{H}}$ decreases as the processes $\mathrm{W}^{(S)}$ get heavier-tailed, Theorem 1 implies that the generalization error can be controlled by the tail behavior of the process: heavier-tails imply less generalization error.

We note that the countability condition on $\mathcal{Z}$ is crucial for Theorem 1. Thanks to this condition, in our proof, we invoke the stability properties of the Hausdorff dimension and we directly obtain a bound on $\dim_{\mathrm{H}} \mathcal{W}$. This bound combined with **H**4 allows us to control the covering number of $\mathcal{W}$, and then the desired result can be obtained by using standard covering techniques [AB09, SSBD14].

Besides, the $\log n$ dependency of $d_\mathrm{H}$ is not crucial; in the supplementary document, we show that the $\log n$ factor can be replaced with any increasing function (e.g, $\log\log n$) by using a chaining argument, with the expense of having $L$ as a multiplying factor (instead of $\log L$). Theorem 1 holds for sufficiently large $n$; however, this threshold is not a-priori known, which is a limitation of the result.

In our second main result, we control the generalization error without the countability assumption on $\mathcal{Z}$, and more importantly we will also relax **H**3. Our main goal will be to relate the error to the Hausdorff dimension of a single $\mathcal{W}_S$, as opposed to $d_\mathrm{H}$, which uniformly bounds $\dim_\mathrm{H}\mathcal{W}_{S'}$ for every $S' \in \mathcal{Z}^n$. In order to achieve this goal, we introduce a technical assumption, which lets us control the statistical dependency between the training set $S$ and the set of parameters $\mathcal{W}_S$.

For any $\delta > 0$, let us consider a finite $\delta$-cover of $\mathcal{W}$ by closed balls of radius $\delta$, whose centers are on the fixed grid $\left\{ \left( \frac{(2j_1+1)\delta}{2\sqrt{d}}, \ldots, \frac{(2j_d+1)\delta}{2\sqrt{d}} \right) : j_i \in \mathbb{Z}, i = 1, \ldots, d \right\}$, and collect the center of each ball in the set $N_\delta$. Then, for each $S$, let us define the set $N_\delta^S := \{ x \in N_\delta : B_d(x, \delta) \cap \mathcal{W}_S \neq \emptyset \}$, where $B_d(x, \delta) \subset \mathbb{R}^d$ denotes the closed ball centered around $x \in \mathbb{R}^d$ with radius $\delta$.

**H 5.** *There exists a constant $M > 1$, such that for any $w \in N_\delta$ and $\epsilon > 0$ we have*

$$\mathbb{P}\left( \{w \in N_\delta^S\} \cap \{|\mathcal{G}(w, S)| \geq \epsilon\} \right) \leq M\mathbb{P}\left( w \in N_\delta^S \right) \mathbb{P}\left( |\mathcal{G}(w, S)| \geq \epsilon \right), \tag{12}$$

*where $\mathcal{G}(w) = \hat{\mathcal{R}}(w, S) - \mathcal{R}(w)$.*

This assumption is common in statistics and is sometimes referred to as the $\psi$-mixing condition, a measure of weak dependence often used in proving limit theorems, see e.g., [Bra83]; yet, it is unfortunately hard to veryify this condition in practice. In our context **H**5 essentially quantifies the dependence between $S$ and the set $\mathcal{W}_S$, through the constant M > 0: smaller $M$ indicates that the dependencce of $\hat{\mathcal{R}}$ on the training sample $S$ is weaker. This concept is also similar to the mutual information used recently in [XR17, AAV18, RZ19] and to the concept of stability [BE02].

**Theorem 2.** *Assume that **H**1, 2 and 5 hold, and **H**4 holds with $\mathcal{W}_S$ in place of $\mathcal{W}$ for all $S \in \mathcal{Z}^n$ (with $a$, $b$, $r_0$, $s$ potentially depending on $S$). Then, for $n$ sufficiently large, we have*

$$\sup_{w \in \mathcal{W}_S} |\hat{\mathcal{R}}(w, S) - \mathcal{R}(w)| \leq B\sqrt{\frac{2\dim_\mathrm{H}\mathcal{W}_S \log(nL^2)}{n} + \frac{\log(4M/\gamma)}{n}}, \tag{13}$$

*with probability at least $1 - \gamma$ over $S \sim \mu_z^{\otimes n}$ and $U \sim \mu_u$.*

This result shows that under **H**5, we can replace $d_\mathrm{H}$ in Theorem 1 with $\dim_\mathrm{H}\mathcal{W}_S$, at the expense of introducing the coupling coefficient $M$ into the bound. We observe that two competing terms are governing the generalization error: in the case where $\dim_\mathrm{H}\mathcal{W}_S$ is small, the error is dominated by the coupling parameter $M$, and vice versa. On the other hand, in the context of Proposition 1, $\dim_\mathrm{H}\mathcal{W}_S \leq \beta_S$, $\mu_u$-almost surely, implying again that a heavy-tailed $\mathrm{W}^{(S)}$ would achieve smaller generalization error as long as the dependency between $S$ and $\mathcal{W}_S$ is weak.

## 4   Experiments

We empirically study the generalization behavior of deep neural networks from the Hausdorff dimension perspective. We use VGG networks [SZ15] as they perform well in practice, and their depth (the number of layers) can be controlled directly. We vary the number of layers from $D = 4$ to $D = 19$, resulting in the number of parameters $d$ between 1.3M and 20M. We train models on the CIFAR-10 dataset [KH09] using SGD and we choose various stepsizes $\eta$, and batch sizes $B$. We provide full range of parameters and additional implementation details in the supplementary document. The code can be found in `https://github.com/umutsimsekli/Hausdorff-Dimension-and-Generalization`.

We assume that SGD can be well-approximated by the process (4). Hence, to bound the corresponding $\dim_\mathrm{H}\mathcal{W}_S$ to be used in Theorem 2, we invoke Proposition 1, which relies on the existence of a point $w_S$ around which the process behaves like a regular Lévy process with exponent $\psi_S$. Considering the empirical observation that SGD exhibits a 'diffusive behavior' around a local minimum [BJSG+18], we take $w_S$ to be the local minimum found by SGD and assume that the conditions of Proposition 1 hold around that point. This perspective indicates that the generalization error can be controlled by the BG index $\beta_S$ of the Lévy process defined by $\psi_S(\xi)$; the sub-symbol of the process (4) around $w_S$.

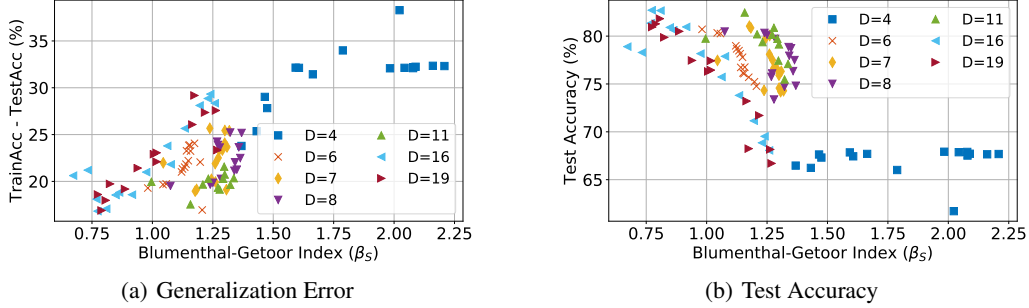

(a) Generalization Error              (b) Test Accuracy

Figure 2: Empirical study of generalization behavior on VGG[SZ15] networks with various depth values (the number of layers are shown as $D$). As our theory predicts, the generalization error is strongly correlated with $\beta_S$. As $\beta_S \in (0, 2]$, the estimates exceeding 2 is an artifact of the estimator.

Estimating the BG index for a general Lévy process is a challenging task; however, the choice of the SDE (4) imposes some structure on $\psi_S$, which lets us express $\beta_S$ in a simpler form. Inspired by the observation that the tail-index of the *gradient noise* in a multi-layer neural network differs from layer to layer, as reported in [ŞGN+19], we will assume that, *around the local minimum* $w_S$, the dynamics of SGD will be similar to the Lévy motion with frozen coefficients: $\Sigma_2(w_S)\mathrm{L}^{\boldsymbol{\alpha}(w_S)}$, see (4) for definitions. We will further impose that, around $w_S$, the coordinates corresponding to the same layer $l$ have the same tail-index $\alpha_l$. Under this assumption, the BG index can be analytically computed as $\beta_S = \max_l \alpha_l \in (0, 2]$ [MX05, Hen73]. While the range $(0, 2]$ might seem narrow at the first sight, we note that $\dim_{\mathrm{H}} \mathcal{W}_S$; hence $\beta_S$ determines the *order* of the generalization error and this parameter gets closer to 0 with more layers added to the network (see Figure 2). Thanks to this simplification, we can easily compute $\beta_S$, by first estimating each $\alpha_l$ by using the estimator proposed in [MMO15], which can efficiently estimate $\alpha_l$ by using multiple SGD iterates.

We trained all the models for 100 epochs and computed their $\beta_S$ over the last epoch, assuming that the iterations reach near local minima. We monitor the generalization error in terms of the difference between the training and test accuracy with respect to the estimated $\beta_S$ in Figure 2(a). We also plot the final test accuracy in Figure 2(b). Test accuracy results validate that the models perform similarly to the state-of-the-art, which suggests that the empirical study matches the practically relevant application settings. Results in Figure 2(a) indicate that, as predicted by our theory, the generalization error is strongly correlated with $\beta_S$, which is an upper-bound of the Hausdorff dimension. With increasing $\beta_S$ (implying increasing Hausdorff dimension), the generalization error increases, as our theory indicates. Moreover, the resulting behavior validates the importance of considering $\dim_{\mathrm{H}} \mathcal{W}_S$ as opposed to ambient Euclidean dimension: for example, the number of parameters in the 4-layer network is significantly lower than other networks; however, its Hausdorff dimension as well as generalization error are significantly higher. Even more importantly, there is no monotonic relationship between the number of parameters and $\dim_{\mathrm{H}} \mathcal{W}_S$. In other words, increasing depth is not always beneficial from the generalization perspective. It is only beneficial if it also decreases $\dim_{\mathrm{H}} \mathcal{W}_S$. We also observe an interesting behavior: the choice of $\eta$ and $B$ seems to affect $\beta_S$, indicating that the choice of the algorithm parameters can impact the tail behavior of the algorithm. In summary, our theory holds over a large selection of depth, step-sizes, and batch sizes when tested on deep neural networks. We provide additional experiments, both real and synthetic, over a collection of model classes in the supplementary document.

## 5 Conclusion

In this paper, we rigorously tied the generalization in a learning task to the tail properties of the underlying training algorithm, shedding light on an empirically observed phenomenon. We established this relationship through the Hausdorff dimension of the SDE approximating the algorithm, and proved a generalization error bound based on this notion of complexity. Unlike the standard ambient dimension, our bounds do not necessarily grow with the number of parameters in the network, and they solely depend on the tail behavior of the training process, providing an explanation for the implicit regularization effect of heavy-tailed SGD.

## Broader Impact

Our work is largely theoretical, studying the generalization properties of deep networks. Our results suggest that the fractal structure and the fractal dimensions of deep learning models can be an accurate metric for the generalization error; hence, in a broader context, we believe that our theory would be useful for practitioners using deep learning tools. On the other hand, our work does not have a direct ethical or societal consequence due to its theoretical nature.

## Acknowledgments and Disclosure of Funding

The authors are grateful to Berfin Şimşek and Xiaochuan Yang for fruitful discussions. The contribution of U.Ş. to this work is partly supported by the French National Research Agency (ANR) as a part of the FBIMATRIX (ANR-16-CE23-0014) project.

## Footnotes

[1]Very recently, Gurbuzbalaban et al. [GSZ20] and Hodgkinson and Mahoney [HM20] have simultaneously shown that the law of the SGD iterates (2) can indeed converge to a heavy-tailed stationary distribution with infinite variance when the step-size $\eta$ is large and/or the batch-size $B$ is small. These results form a theoretical basis for the origins of the observed heavy-tailed behavior of SGD in practice.

[2]Here $\mathrm{L}_t^\alpha$ is equivalent to $\mathrm{L}_t^{\boldsymbol{\alpha}(\cdot)}$ with $\alpha_i(w) = \alpha \in (0,2], \forall i \in \{1,\ldots,d\}$ and $\forall w \in \mathbb{R}^d$.

[3]Note that this prevents adaptive minibatching algorithms e.g., [AB99] to be represented in our framework.

[4]**H**4 ensures that the Hausdorff dimension of $\mathcal{W}$ coincides with another notion of dimension, called the Minkowski dimension, which is explained in detail in the supplement. We note that for many fractal-like sets, these two notions of dimensions are equal to each other (see [Mat99] Chapter 5), which include $\alpha$-stable processes (see [Fal04] Chapter16).

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
