[Supplementary Material]

# Hausdorff Dimension, Heavy Tails, and Generalization in Neural Networks

## SUPPLEMENTARY DOCUMENT

**Umut Şimşekli[1,2], Ozan Sener[3], George Deligiannidis[2,4], Murat A. Erdogdu[5,6]**
LTCI, Télécom Paris, Institut Polytechnique de Paris[1], University of Oxford[2], Intel Labs[3]
The Alan Turing Institute[4], University of Toronto[5], Vector Institute[6]

## Contents

## S1    Additional Experimental Results and Implementation Details

### S1.1    Comparison with Other Generalization Metrics for Deep Networks

In this section, we empirically analyze the proposed metric with respect to existing generalization metrics, developed for neural networks. Specifically, we consider the 'flat minima' argument of Jastrzevski et al. [JKA$^+$17] and plot the generalization error vs $\eta/B$ which is the ratio of step size to the batch size. As a second comparison, we use heavy-tailed random matrix theory based metric of Martin and Mahoney [MM19]. We plot the generalization error with respect to each metric in Figure S1. As the results suggest, our metric is the one which correlates best with the empirically

observed generalization error. The metric proposed by Martin and Mahoney [MM19] fails for the low number of layers and the resulting behavior is not monotonic. Similarly, $\eta/B$ captures the relationship for very deep networks (for $D = 16\&19$), however, it fails for other settings.

We also note that the norm-based capacity metrics [NTS15] typically increase with the increasing dimension $d$, we refer to [NBMS17] for details.

(a) Hausdorff Dimension [Ours]     (c) $\alpha$ [MM19]     (b) $\eta/B$ [JKA$^+$17]

Figure S1: Empirical comparison to other capacity metrics.

## S1.2 Synthetic Experiments

We consider a simple synthetic logistic regression problem, where the data distribution is a Gaussian mixture model with two components. Each data point $z_i \equiv (x_i, y_i) \in \mathcal{Z} = \mathbb{R}^d \times \{-1, 1\}$ is generated by simulating the model: $y_i \sim \mathrm{Bernoulli}(1/2)$ and $x_i|y_i \sim \mathcal{N}(m_{y_i}, 100\mathrm{I}_d)$, where the means are drawn from a Gaussian: $m_{-1}, m_1 \sim \mathcal{N}(0, 25\mathrm{I}_d)$. The loss function $\ell$ is the logistic loss as $\ell(w, z) = \log(1 + \exp(-yx^\top w))$.

Figure S2: Results on synthetic data.

As for the algorithm, we consider a data-independent multivariate stable process: $[\mathcal{A}(S)]_t = \mathrm{L}_t^\alpha$ for any $S \in \mathcal{Z}^n$, where $\mathrm{L}_1^\alpha$ is distributed with an elliptically contoured $\alpha$-stable distribution with $\alpha \in (0, 2]$ (see Section 2): when $\alpha = 2$, $\mathrm{L}_t^\alpha$ is just a Brownian motion, as $\alpha$ gets smaller, the process becomes heavier-tailed. By Theorem 4.2 of [BG60], $\mathcal{A}$ has the uniform Hausdorff dimension property with $d_\mathrm{H} = \alpha$ independently from $d$ when $d \geq 2$.

We set $d = 10$ and generate points to represent the whole population, i.e., $\{z_i\}_{i=1}^{n_{\mathrm{tot}}}$ with $n_{\mathrm{tot}} = 100\mathrm{K}$. Then, for different values of $\alpha$, we simulate $\mathcal{A}$ for $t \in [0, 1]$, by using a small step-size $\eta = 0.001$ (the total number of iterations is hence $1/\eta$). We finally draw 20 random sets $S$ with $n$ elements from this population, and we monitor the maximum difference $\sup_{w \in \mathcal{W}_S} |\hat{\mathcal{R}}(w, S) - \mathcal{R}(w)|$ for different values of $n$. We repeat the whole procedure 20 times and report the average values in Figure S2. We observe that the results support Theorems 1 and 2: for every $n$, the generalization error decreases with decreasing $\alpha$, hence illustrates the role of the Hausdorff dimension.

## S1.3 Implementation Details for the Deep Neural Network Experiments

In this section, we provide the additional details which are skipped in the main text for the sake of space. We use the following VGG-style neural networks with various number of layers as

- **VGG4**: Conv(512) - ReLU - MaxPool - Linear

- **VGG6**: Conv(256) - ReLU - MaxPool - Conv(512) - ReLU - MaxPool - Conv(512) - ReLU - MaxPool - Linear

- **VGG7**: Conv(128) - ReLU - MaxPool - Conv(256) - ReLU - MaxPool - Conv(512) - ReLU - MaxPool - Conv(512) - ReLU - MaxPool - Linear

- **VGG8**: Conv(64) - ReLU - MaxPool - Conv(128) - ReLU - MaxPool - Conv(256) - ReLU - MaxPool - Conv(512) - ReLU - MaxPool - Conv(512) - ReLU - MaxPool - Linear

- **VGG11**: Conv(64) - ReLU - MaxPool - Conv(128) - ReLU - MaxPool - Conv(256) - ReLU - Conv(256) - ReLU - MaxPool - Conv(512) - ReLU - Conv(512) - ReLU - MaxPool - Conv(512) - ReLU - Conv(512) - ReLU - MaxPool - Linear

- **VGG16**: Conv(64) - ReLU - Conv(64) - ReLU - MaxPool - Conv(128) - ReLU - Conv(128) - ReLU - MaxPool - Conv(256) - ReLU - Conv(256) - ReLU - Conv(256) - ReLU - MaxPool - Conv(512) - ReLU - Conv(512) - ReLU - Conv(512) - ReLU - MaxPool - Conv(512) - ReLU - Conv(512) - ReLU - Conv(512) - ReLU - MaxPool - Linear

- **VGG19**: Conv(64) - ReLU - Conv(64) - ReLU - MaxPool - Conv(128) - ReLU - Conv(128) - ReLU - MaxPool - Conv(256) - ReLU - Conv(256) - ReLU - Conv(256) - ReLU - Conv(256) - ReLU - MaxPool - Conv(512) - ReLU - Conv(512) - ReLU - Conv(512) - ReLU - Conv(512) - ReLU - MaxPool - Conv(512) - ReLU - Conv(512) - ReLU - Conv(512) - ReLU - Conv(512) - ReLU - MaxPool - Linear

where all convolutions are noted with the number of filters in the paranthesis. Moreover, we use the following hyperparameter ranges for step size of SGD: $\{1e-2, 1e-3, 3e-3, 1e-4, 3e-4, 1e-5, 3e-5\}$ with the batch sizes $\{32, 64, 128, 256\}$. All networks are learned with cross entropy loss and ReLU activations, and no additional technique like batch normalization or dropout is used. We will also release the full source code of the experiments.

## S2  Representing Optimization Algorithms as Feller Processes

Thanks to the generality of the Feller processes, we can represent multiple popular stochastic optimization algorithms as a Feller process, in addition to SGD. For instance, let us consider the following SDE:

$$\mathrm{dW}_t = -\Sigma_0(\mathrm{W}_t)\nabla f(\mathrm{W}_t)\mathrm{d}t + \Sigma_1(\mathrm{W}_t)\mathrm{dB}_t + \Sigma_2(\mathrm{W}_t)\mathrm{dL}_t^{\boldsymbol{\alpha}(\mathrm{W}_t)}, \tag{S1}$$

where $\Sigma_0, \Sigma_1, \Sigma_2$ are $d \times d$ matrix-valued functions and the tail-index $\boldsymbol{\alpha}(\cdot)$ of $\mathrm{L}_t^{\boldsymbol{\alpha}}(\cdot)$ is also allowed to change depending on value of the state $\mathrm{W}_t$. We can verify that this SDE corresponds to a Feller process with $b(w) = -\Sigma_0(w)\nabla f(w)$, $\Sigma(w) = 2\Sigma_1(w)$, and an appropriate choice of $\nu$ [HDS18]. As we discussed in the main document, we the choice $\Sigma_0 = \mathrm{I}_d$ can represent SGD with state-dependent Gaussian and/or heavy-tailed noise. Besides, we can choose an appropriate $\Sigma_0$ in order to be able to represent optimization algorithms that use second-order geometric information, such as natural gradient [Ama98] or stochastic Newton [EM15] algorithms. On the other hand, by using the SDEs proposed in [GGZ18, ŞZTG20, LPH+17, OKL19, BB18], we can further represent momentum-based algorithms such as SGD with momentum [Pol64] as a Feller process.

## S3  Decomposable Feller Processes and their Hausdorff Dimension

In our study, we focus on decomposable Feller processes, introduced in [Sch98]. Let us consider a Feller process expressed by its symbol $\Psi$. We call the process defined by $\Psi$ decomposable at $w_0$, if there exists a point $w_0 \in \mathbb{R}^d$ such that the symbol can be decomposed as

$$\Psi(w, \xi) = \psi(\xi) + \tilde{\Psi}(w, \xi), \tag{S2}$$

where $\psi(\xi) := \Psi(w_0, \xi)$ is the sub-symbol and $\tilde{\Psi}(w, \xi) = \Psi(w, \xi) - \Psi(w_0, \xi)$ is the reminder term. Let $\mathbf{j} \in \mathbb{N}_0^d$ denote a multi-index[1]. We assume that there exist functions $a, \Phi_{\mathbf{j}} : \mathbb{R}^d \mapsto \mathbb{R}$ such that the following holds:

- $\Psi(x, 0) \equiv 0$
- $\|\Phi_0\|_\infty < \infty$, and $\Phi_{\mathbf{j}} \in L^1\left(\mathbb{R}^d\right)$ for all $|\mathbf{j}| \leq d+1$.
- $\left|\partial_w^{\mathbf{j}} \tilde{\Psi}(w, \xi)\right| \leq \Phi_{\mathbf{j}}(w)\left(1 + a^2(\xi)\right)$, for all $w, \xi \in \mathbb{R}^d$ and $|\mathbf{j}| \leq d+1$.
- $a^2(\xi) \geq \kappa_0 \|\xi\|^{r_0}$, for $\|\xi\|$ large, $r_0 \in (0, 2]$, and $\kappa_0 > 0$.

The Hausdorff dimension of the image of a decomposable Feller process is bounded, due to the following result.

**Theorem S1** ([Sch98] Theorem 4). *Let $\Psi(x, \xi)$ generate a Feller process, denoted by $\{W_t\}_{t\geq 0}$. Assume that $\Psi$ is decomposable at $w_0$ with the sub-symbol $\psi$. Then, for any given $T \in \mathbb{R}_+$, we have*

$$\dim_H W([0, T]) \leq \beta, \qquad \mathbb{P}^x\text{-almost surely,} \tag{S3}$$

*where $W([0, T]) := \{w : w = W_t, \text{ for some } t \in [0, T]\}$ is the image of the process, $\mathbb{P}^x$ denotes the law of the process $\{W_t\}_{t\geq 0}$ with initial value $x$, and $\beta$ is the upper Blumenthal-Getoor index of the Lévy process with the characteristic exponent $\psi(\xi)$, given as follows:*

$$\beta := \inf\left\{\lambda \geq 0 : \lim_{\|\xi\|\to\infty} \frac{|\psi(\xi)|}{\|\xi\|^\lambda} = 0\right\}. \tag{S4}$$

## S4 Improving the Convergence Rate via Chaining

In this section, we present additional theoretical results. We improve the bound in Theorem 1, in the sense that we replace the $\log n$ factor any increasing function.

**Theorem S2.** *Assume that H1 to 4 hold, and $\mathcal{Z}$ is countable. Then, for any function $\xi : \mathbb{R} \to \mathbb{R}$ satisfying $\lim_{x\to\infty} \rho(x) = \infty$, and for a sufficiently large $n$, we have*

$$\sup_{w \in \mathcal{W}_S}\left(\hat{\mathcal{R}}(w, S) - \mathcal{R}(w)\right) \leq cLB\text{diam}(\mathcal{W})\sqrt{\frac{d_H\rho(n)}{n} + \frac{\log(1/\gamma)}{n}},$$

*with probability at least $1 - \gamma$ over $S \sim \mu_z^{\otimes n}$ and $U \sim \mu_u$, where $c$ is an absolute constant.*

This result implies that the $\log n$ dependency of Theorem 1 can be replaced with any increasing function $\rho$, at the expense of having the constant $\text{diam}(\mathcal{W})$ and having $L$ instead of $\log L$ in the bound.

## S5 Additional Technical Background

In this section, we will define the notions that will be used in our proofs. For the sake of completeness we also provide the main theoretical results that will be used in our proofs.

### S5.1 The Minkowski Dimension

In our proofs, in addition to the Hausdorff dimension, we also make use of another notion of dimension, referred to as the Minkowski dimension (also known as the box-counting dimension [Fal04]), which is defined as follows.

**Definition S1.** *Let $G \subset \mathbb{R}^d$ be a set and let $N_\delta(G)$ be a collection of sets that contains either one of the following:*

- *The smallest number of sets of diameter at most $\delta$ which cover $G$*

- *The smallest number of closed balls of diameter at most $\delta$ which cover $G$*

- *The smallest number of cubes of side at most $\delta$ which cover $G$*

- *The number of $\delta$-mesh cubes that intersect $G$*

- *The largest number of disjoint balls of radius $\delta$, whose centers are in $G$.*

*Then the lower- and upper-Minkowski dimensions of G are respectively defined as follows:*

$$\underline{\dim}_{\mathrm{M}}G := \liminf_{\delta\to 0}\frac{\log|N_\delta(G)|}{-\log\delta}, \qquad \overline{\dim}_{\mathrm{M}}G := \limsup_{\delta\to 0}\frac{\log|N_\delta(G)|}{-\log\delta}. \qquad \text{(S5)}$$

*In case the* $\underline{\dim}_{\mathrm{M}}G = \overline{\dim}_{\mathrm{M}}G$, *the Minkowski dimension* $\dim_{\mathrm{M}}(G)$ *is their common value.*

We always have $0 \le \dim_{\mathrm{H}} G \le \underline{\dim}_{\mathrm{M}}G \le \overline{\dim}_{\mathrm{M}}G \le d$ where the inequalities can be strict [Fal04].

It is possible to construct examples where the Hausdorff and Minkowski dimensions are different from each other. However, in many interesting cases, these two dimensions often match each other [Fal04]. In this paper, we are interested in such a case, i.e. the case when the Hausdorff and Minkowski dimensions match. The following result identifies the conditions for which the two dimensions match each other, which form the basis of **H**4:

**Theorem S3** ([Mat99] Theorem 5.7). *Let $A$ be a non-empty bounded subset of $\mathbb{R}^d$. Suppose there is a Borel measure $\mu$ on $\mathbb{R}^d$ and there are positive numbers $a, b, r_0$ and $s$ such that $0 < \mu(A) \le \mu\left(\mathbb{R}^d\right) < \infty$ and*

$$0 < ar^s \le \mu(B_d(x,r)) \le br^s < \infty \quad for \quad x \in A, 0 < r \le r_0 \qquad \text{(S6)}$$

*Then* $\dim_{\mathrm{H}} A = \dim_{\mathrm{M}} A = \overline{\dim}_{\mathrm{M}}A = s$.

### S5.2 Egoroff's Theorem

Egoroff's theorem is an important result in measure theory and establishes a condition for measurable functions to be uniformly continuous in an almost full-measure set.

**Theorem S4** (Egoroff's Theorem [Bog07] Theorem 2.2.1). *Let $(X, \mathcal{A}, \mu)$ be a space with a finite nonnegative measure $\mu$ and let $\mu$-measurable functions $f_n$ be such that $\mu$-almost everywhere there is a finite limit $f(x) := \lim_{n\to\infty} f_n(x)$. Then, for every $\varepsilon > 0$, there exists a set $X_\varepsilon \in \mathcal{A}$ such that $\mu\left(X\backslash X_\varepsilon\right) < \varepsilon$ and the functions $f_n$ converge to $f$ uniformly on $X_\varepsilon$.*

## S6 Postponed Proofs

### S6.1 Proof of Proposition 1

*Proof.* Let $\Psi_S$ denote the symbol of the process $\mathrm{W}^{(S)}$. Then, the desired result can obtained by directly applying Theorem S1 on each $\Psi_S$. ☐

### S6.2 Proof of Theorem 1

We first prove the following more general result which relies on $\overline{\dim}_{\mathrm{M}}\mathcal{W}$.

**Lemma S1.** *Assume that $\ell$ is bounded by $B$ and $L$-Lipschitz continuous in $w$. Let $\mathcal{W} \subset \mathbb{R}^d$ be a set with finite diameter. Then, for $n$ sufficiently large, we have*

$$\sup_{w\in\mathcal{W}}|\hat{\mathcal{R}}(w,S) - \mathcal{R}(w)| \le B\sqrt{\frac{2\overline{\dim}_{\mathrm{M}}\mathcal{W}\log(nL^2)}{n} + \frac{\log(1/\gamma)}{n}}, \qquad \text{(S7)}$$

*with probability at least $1 - \gamma$ over $S \sim \mu_z^{\otimes n}$.*

*Proof.* As $\ell$ is L-Lipschitz, so are $\mathcal{R}$ and $\hat{\mathcal{R}}$. By using the notation $\hat{\mathcal{R}}_n(w) := \hat{\mathcal{R}}(w,S)$, and by the triangle inequality, for any $w' \in \mathcal{W}$ we have:

$$|\hat{\mathcal{R}}_n(w) - \mathcal{R}(w)| = \left|\hat{\mathcal{R}}_n\left(w'\right) - \mathcal{R}\left(w'\right) + \hat{\mathcal{R}}_n(w) - \hat{\mathcal{R}}_n\left(w'\right) - \mathcal{R}(w) + \mathcal{R}\left(w'\right)\right| \qquad \text{(S8)}$$

$$\le \left|\hat{\mathcal{R}}_n\left(w'\right) - \mathcal{R}\left(w'\right)\right| + 2L\left\|w - w'\right\|. \qquad \text{(S9)}$$

Now since $\mathcal{W}$ has finite diameter, let us consider a finite $\delta$-cover of $\mathcal{W}$ by balls and collect the center of each ball in the set $N_\delta := N_\delta(\mathcal{W})$. Then, for each $w \in \mathcal{W}$, there exists a $w' \in N_\delta$, such that $\|w - w'\| \le \delta$. By choosing this $w'$ in the above inequality, we obtain:

$$|\hat{\mathcal{R}}_n(w) - \mathcal{R}(w)| \le \left|\hat{\mathcal{R}}_n\left(w'\right) - \mathcal{R}\left(w'\right)\right| + 2L\delta. \qquad \text{(S10)}$$

Taking the supremum of both sides of the above equation yields:

$$\sup_{w \in \mathcal{W}} |\hat{\mathcal{R}}_n(w) - \mathcal{R}(w)| \leq \max_{w \in N_\delta} \left| \hat{\mathcal{R}}_n(w) - \mathcal{R}(w) \right| + 2L\delta. \tag{S11}$$

Using the union bound over $N_\delta$, we obtain

$$\mathbb{P}_S \left( \max_{w \in N_\delta} |\hat{\mathcal{R}}_n(w) - \mathcal{R}(w)| \geq \varepsilon \right) = \mathbb{P}_S \left( \bigcup_{w \in N_\delta} \left\{ |\hat{\mathcal{R}}_n(w) - \mathcal{R}(w)| \geq \varepsilon \right\} \right) \tag{S12}$$

$$\leq \sum_{w \in N_\delta} \mathbb{P}_S \left( |\hat{\mathcal{R}}_n(w) - \mathcal{R}(w)| \geq \varepsilon \right). \tag{S13}$$

Further, for $\delta > 0$, since $|N_\delta|$ has finitely many elements, we can invoke Hoeffding's inequality for each of the summands on the right hand side and obtain

$$\mathbb{P}_S \left( \max_{w \in N_\delta} |\hat{\mathcal{R}}_n(w) - \mathcal{R}(w)| \geq \varepsilon \right) \leq 2|N_\delta| \exp \left\{ -\frac{2n\varepsilon^2}{B^2} \right\} =: \gamma. \tag{S14}$$

Notice that $N_\delta$ is a random set, and choosing $\varepsilon$ based on $|N_\delta|$, one can obtain a deterministic $\gamma$. Therefore, we can plug this back in (S11) and obtain that, with probability at least $1 - \gamma$

$$\sup_{w \in \mathcal{W}} |\hat{\mathcal{R}}_n(w) - \mathcal{R}(w)| \leq B \sqrt{\frac{\log(2|N_\delta|)}{2n} + \frac{\log(1/\gamma)}{2n}} + 2L\delta. \tag{S15}$$

Now since $\mathcal{W} \subset \mathbb{R}^d$, $\overline{\dim_{\mathrm{M}}}\mathcal{W}$ is finite. Then, for any sequence $\{\delta_n\}_{n \in \mathbb{N}}$ such that $\lim_{n \to \infty} \delta_n = 0$, we have, $\forall \epsilon > 0$, $\exists n_\epsilon > 0$ such that $n \geq n_\epsilon$ implies

$$\log(|N_\delta|) \leq (\overline{\dim_{\mathrm{M}}}\mathcal{W} + \epsilon) \log(\delta_n^{-1}). \tag{S16}$$

Choosing $\delta_n = 1/\sqrt{nL^2}$ and $\epsilon = \overline{\dim_{\mathrm{M}}}\mathcal{W}$, we have for $\forall n \geq n_{\overline{\dim_{\mathrm{M}}}\mathcal{W}}$,

$$\log(2|N_\delta|) \leq \log(2) + \overline{\dim_{\mathrm{M}}}\mathcal{W} \log(nL^2) \quad \text{and} \quad 2L\delta_n = \frac{2}{\sqrt{n}}. \tag{S17}$$

Therefore, we obtain with probability at least $1 - \gamma$

$$\sup_{w \in \mathcal{W}} |\hat{\mathcal{R}}_n(w) - \mathcal{R}(w)| \leq B \sqrt{\frac{\log(2) + \overline{\dim_{\mathrm{M}}}\mathcal{W} \log(nL^2)}{2n} + \frac{\log(1/\gamma)}{2n}} + \frac{2}{\sqrt{n}}, \tag{S18}$$

$$\leq B \sqrt{\frac{2\overline{\dim_{\mathrm{M}}}\mathcal{W} \log(nL^2)}{n} + \frac{\log(1/\gamma)}{n}}, \tag{S19}$$

for sufficiently large $n$. This concludes the proof. $\qquad\square$

We now proceed to the proof of Theorem 1.

*Proof of Theorem 1.* By noticing $\mathcal{Z}^n$ is countable (since $\mathcal{Z}$ is countable) and using the property that $\dim_{\mathrm{H}} \cup_{i \in \mathbb{N}} A_i = \sup_{i \in \mathbb{N}} \dim_{\mathrm{H}} A_i$ (cf. [Fal04], Section 3.2), we observe that

$$\dim_{\mathrm{H}} \mathcal{W} = \dim_{\mathrm{H}} \bigcup_{S \in \mathcal{Z}^n} \mathcal{W}_S = \sup_{S \in \mathcal{Z}^n} \dim_{\mathrm{H}} \mathcal{W}_S \leq d_{\mathrm{H}}, \tag{S20}$$

$\mu_u$-almost surely. Define the event $\mathcal{Q}_R = \{\mathrm{diam}(\mathcal{W}) \leq R\}$. On the event $\mathcal{Q}_R$, by Theorem S3, we have that $\dim_{\mathrm{M}} \mathcal{W} = \overline{\dim_{\mathrm{M}}}\mathcal{W} = \dim_{\mathrm{H}} \mathcal{W} \leq d_{\mathrm{H}}$, $\mu_u$-almost surely.

Now, we observe that

$$\sup_{w \in \mathcal{W}_S} |\hat{\mathcal{R}}(w, S) - \mathcal{R}(w)| \leq \sup_{w \in \mathcal{W}} |\hat{\mathcal{R}}(w, S) - \mathcal{R}(w)|. \tag{S21}$$

Hence, by defining $\varepsilon = B\sqrt{\frac{2(\overline{\dim_{\mathrm{M}}}\mathcal{W})\log(nL^2)}{n} + \frac{\log(1/\gamma)}{n}}$, and using the independence of $S$ and $U$, Lemma S1, and (S20), we write

$$
\mathbb{P}_{S,U}\left(\sup_{w\in\mathcal{W}_S}|\hat{\mathcal{R}}(w,S)-\mathcal{R}(w)| > B\sqrt{\frac{2d_{\mathrm{H}}\log(nL^2)}{n} + \frac{\log(1/\gamma)}{n}}; \mathcal{Q}_R\right)
$$

$$
\leq \mathbb{P}_{S,U}\left(\sup_{w\in\mathcal{W}}|\hat{\mathcal{R}}(w,S)-\mathcal{R}(w)| > B\sqrt{\frac{2d_{\mathrm{H}}\log(nL^2)}{n} + \frac{\log(1/\gamma)}{n}}; \mathcal{Q}_R\right)
$$

$$
= \mathbb{P}_{S,U}\left(\sup_{w\in\mathcal{W}}|\hat{\mathcal{R}}(w,S)-\mathcal{R}(w)| > B\sqrt{\frac{2d_{\mathrm{H}}\log(nL^2)}{n} + \frac{\log(1/\gamma)}{n}} \; ; \; \overline{\dim_{\mathrm{M}}}\mathcal{W} \leq d_{\mathrm{H}}; \mathcal{Q}_R\right)
$$

$$
\leq \mathbb{P}_{S,U}\left(\sup_{w\in\mathcal{W}}|\hat{\mathcal{R}}(w,S)-\mathcal{R}(w)| > \varepsilon \; ; \mathcal{Q}_R\right).
$$

Finally, we let $R\to\infty$ and use dominated convergence theorem to obtain

$$
\mathbb{P}_{S,U}\left(\sup_{w\in\mathcal{W}_S}|\hat{\mathcal{R}}(w,S)-\mathcal{R}(w)| > B\sqrt{\frac{2d_{\mathrm{H}}\log(nL^2)}{n} + \frac{\log(1/\gamma)}{n}}\right)
$$

$$
\leq \mathbb{P}_{S,U}\left(\sup_{w\in\mathcal{W}}|\hat{\mathcal{R}}(w,S)-\mathcal{R}(w)| > \varepsilon\right)
$$

$$
\leq \gamma,
$$

which concludes the proof. $\qquad\square$

### S6.3 Proof of Theorem 2

*Proof.* Due to the hypotheses and Theorem S3, we have $\overline{\dim_{\mathrm{M}}}\mathcal{W}_S = \dim_{\mathrm{H}}\mathcal{W}_S := d_{\mathrm{H}}$, $\mu_u$-almost surely. Note that in this setting, $d_{\mathrm{H}}$ denotes the dimension of a specific $\mathcal{W}_S$ and can depend on $S$.

It is easy to verify that the particular forms of the $\delta$-covers and $N_\delta$ in **H**5 still yield the same Minkowski dimension in (S5). Then by definition, we have for all $S$:

$$
\limsup_{\delta\to 0}\frac{\log|N_\delta^S|}{\log(1/\delta)} = \limsup_{\delta\to 0}\sup_{r<\delta}\frac{\log|N_r^S|}{\log(1/r)} = d_{\mathrm{H}},
\tag{S22}
$$

$\mu_u$-almost surely. By applying Theorem S4 to the collection of random variables:

$$
f_r(S) := \sup_{r<\delta}\frac{\log|N_r^S|}{\log(1/r)},
\tag{S23}
$$

for any $\delta' > 0$ we can find a subset $\mathfrak{Z} \subset \mathcal{Z}^n$, with probability at least $1 - \delta'$, such that on $\mathfrak{Z}$ the convergence is uniform. That is for any $S \in \mathfrak{Z}$ and any $\epsilon'$, there is a $\delta_0 = \delta(\epsilon')$ such that for all $\delta < \delta_0$ we have $f_\delta(S) < d_{\mathrm{M}} + \epsilon'$.

As $U$ and $S$ are assumed to be independent, all the following statements hold $\mu_u$-almost surely, hence we drop the dependence on $U$ to ease the notation. We proceed as in the proof of Lemma S1:

$$
\sup_{w\in\mathcal{W}_S}|\hat{\mathcal{R}}_n(w)-\mathcal{R}(w)| \leq \max_{w\in N_\delta^S}\left|\hat{\mathcal{R}}_n(w)-\mathcal{R}(w)\right| + 2L\delta.
\tag{S24}
$$

Using the union bound over $N_\delta^S$, we obtain

$$
\mathbb{P}_S\left(\max_{w\in N_\delta^S}|\hat{\mathcal{R}}_n(w)-\mathcal{R}(w)| \geq \varepsilon\right)
\tag{S25}
$$

$$
\leq \sum_{w\in N_\delta}\mathbb{P}_S\left(\left\{|\hat{\mathcal{R}}_n(w)-\mathcal{R}(w)| \geq \varepsilon\right\}\cap\{w\in N_\delta^S\}\right).
\tag{S26}
$$

Continuing from above and applying Assumption 5 we get

$$\mathbb{P}_S\left(\max_{w \in N_\delta^S} |\hat{\mathcal{R}}_n(w) - \mathcal{R}(w)| \geq \varepsilon\right) \tag{S27}$$

$$\leq M \sum_{w \in N_\delta} \mathbb{P}_S\left(w \in N_\delta^S\right) \times \mathbb{P}_S\left(|\hat{\mathcal{R}}_n(w) - \mathcal{R}(w)| \geq \varepsilon\right) \tag{S28}$$

$$\leq M \sum_{w \in N_\delta} \left[\mathbb{P}_S\left(\mathfrak{Z} \cap \{w \in N_\delta^S\}\right) + \delta'\right] \times \mathbb{P}_S\left(|\hat{\mathcal{R}}_n(w) - \mathcal{R}(w)| \geq \varepsilon\right) \tag{S29}$$

and applying Hoeffding's inequality for each of the summands on the right hand side and obtain

$$= 2M \exp\left\{-\frac{2n\varepsilon^2}{B^2}\right\} \sum_{w \in N_\delta} \left(\mathbb{E}_S\left[\mathbb{1}\{S \in \mathfrak{Z}\}\mathbb{1}\{w \in N_\delta^S\}\right] + \delta'\right) \tag{S30}$$

$$= 2M \exp\left\{-\frac{2n\varepsilon^2}{B^2}\right\} \left(\mathbb{E}_S\left[\mathbb{1}\{S \in \mathfrak{Z}\}|N_\delta^S|\right] + |N_\delta|\delta'\right), \tag{S31}$$

where $\mathbb{1}$ denotes the indicator function.

At this point, using Theorem S4 as explained above, for any $\epsilon'$ there exists $\delta_0 = \delta_0(\epsilon')$, such that for any $\delta < \delta_0$ and $S \in \mathfrak{Z}$ we have $|N_\delta(S)| \leq (1/\delta)^{d_H + \epsilon'}$. In addition we know that $|N_\delta| \leq (3\mathrm{diam}(\mathcal{W})/\delta)^d$. Therefore we have

$$\mathbb{P}_S\left(\max_{w \in N_\delta^S} |\hat{\mathcal{R}}_n(w) - \mathcal{R}(w)| \geq \varepsilon\right) \tag{S32}$$

$$\leq 2M \exp\left\{-\frac{2n\varepsilon^2}{B^2}\right\} \left[\left(\frac{1}{\delta}\right)^{d_H + \epsilon'} + \delta'\left(\frac{3\mathrm{diam}(\mathcal{W})}{\delta}\right)^d\right]. \tag{S33}$$

At this point set

$$\delta' := \frac{(1/\delta)^{d_H + \epsilon'}}{(3\mathrm{diam}(\mathcal{W})/\delta)^d},$$

so that

$$\mathbb{P}_S\left(\max_{w \in N_\delta^S} |\hat{\mathcal{R}}_n(w) - \mathcal{R}(w)| \geq \varepsilon\right) \leq 4M \exp\left\{-\frac{2n\varepsilon^2}{B^2}\right\} \left(\frac{1}{\delta}\right)^{d_H + \epsilon'} =: \gamma. \tag{S34}$$

Therefore with probability at least $1 - \gamma$ we have

$$\sup_{w \in \mathcal{W}_S} |\hat{\mathcal{R}}_n(w) - \mathcal{R}(w)| \leq B\sqrt{\frac{(d_H + \epsilon')\log(1/\delta) + \log 4 + \log M + \log(1/\gamma)}{2n}} + 2L\delta. \tag{S35}$$

Choosing $\delta_n = 1/\sqrt{nL^2}$ and $\epsilon' = d_H$, we have for all $n \geq n_0$,

$$\sup_{w \in \mathcal{W}_S} |\hat{\mathcal{R}}_n(w) - \mathcal{R}(w)| \leq B\sqrt{\frac{2d_H \log(nL^2) + \log(4M/\gamma)}{n}}, \tag{S36}$$

for sufficiently large $n_0$. This concludes the proof. $\square$

### S6.4 Proof of Theorem S2

Similar to the proof of Theorem 1, we first prove a more general result where $\overline{\dim_M}\mathcal{W}$ is fixed.

**Lemma S2.** *Assume that $\ell$ is bounded by $B$ and $L$-Lipschitz continuous in $w$. Let $\mathcal{W} \subset \mathbb{R}^d$ be a bounded set with $\overline{\dim_M}\mathcal{W} \leq d_M$. For any function $\rho : \mathbb{R} \to \mathbb{R}$ satisfying $\lim_{x \to \infty} \rho(x) = \infty$ and for a sufficiently large $n$, with probability at least $1 - \gamma$, we have*

$$\sup_{w \in \mathcal{W}} \left(\hat{\mathcal{R}}_n(w) - \mathcal{R}(w)\right) \leq cLB\mathrm{diam}(\mathcal{W})\sqrt{\frac{d_M\rho(n) + \log(1/\gamma)}{n}},$$

*where $c$ is an absolute constant.*

*Proof.* We define the empirical process

$$\mathcal{G}_n(w) := \hat{\mathcal{R}}_n(w) - \mathcal{R}(w) = \frac{1}{n}\sum_{i=1}^n \ell(w, z_i) - \mathbb{E}_z[\ell(w, z)],$$

and we notice that

$$\mathbb{E}[\mathcal{G}_n(w)] = 0.$$

Recall that a random process $\{G(w)\}_{w \in \mathcal{W}}$ on a metric space $(\mathcal{W}, d)$ is said to have sub-Gaussian increments if there exists $K \geq 0$ such that

$$\|G(w) - G(w')\|_{\psi_2} \leq Kd(w, w'), \tag{S37}$$

where $\|\cdot\|_{\psi_2}$ denotes the sub-Gaussian norm [Ver19].

We verify that $\{\mathcal{G}_n(w)\}_w$ has sub-Gaussian increments with $K = 2L/\sqrt{n}$ and for the metric being the standard Euclidean metric, $d(w, w') = \|w - w'\|$. To see why this is the case, notice that

$$\mathcal{G}_n(w) - \mathcal{G}_n(w') = \frac{1}{n}\sum_{i=1}^n [\ell(w, z_i) - \ell(w', z_i) - (\mathbb{E}_z\ell(w, z) - \mathbb{E}_z\ell(w', z))]$$

which is a sum of i.i.d. random variables that are uniformly bounded by

$$|\ell(w, z_i) - \ell(w', z_i) - (\mathbb{E}_z\ell(w, z) - \mathbb{E}_z\ell(w', z))| \leq 2L\|w - w'\|,$$

by the Lipschitz continuity of the loss. Therefore, Hoeffding's lemma for bounded and centered random variables easily imply that

$$\mathbb{E}\left\{\exp\left[\lambda\left(\mathcal{G}_n(w) - \mathcal{G}_n(w')\right)\right]\right\} \leq \exp\left[\frac{2\lambda^2}{n}L^2\|w - w'\|^2\right], \tag{S38}$$

thus, we have $\|\mathcal{G}_n(w) - \mathcal{G}_n(w')\|_{\psi_2} \leq (2L/\sqrt{n})\|w - w'\|$.

Next, define the sequence $\delta_k = 2^{-k}$ and notice that we have $\delta_k \downarrow 0$. Dudley's tail bound (see for example [Ver19, Thm. 8.1.6]) for this empirical process implies that, with probability at least $1 - \gamma$, we have

$$\sup_{w,w' \in \mathcal{W}} (\mathcal{G}_n(w) - \mathcal{G}_n(w')) \leq C\frac{L}{\sqrt{n}}\left[S_{\mathcal{W}} + \sqrt{\log(2/\gamma)}\mathrm{diam}(\mathcal{W})\right] \tag{S39}$$

where $C$ is an absolute constant and

$$S_{\mathcal{W}} = \sum_{k \in \mathbb{Z}} \delta_k \sqrt{\log|N_{\delta_k}(\mathcal{W})|}.$$

In order to apply Dudley's lemma, we need to bound the above summation. For that, choose $\kappa_0$ such that

$$2^{\kappa_0} \geq \mathrm{diam}(\mathcal{W}) > 2^{\kappa_0 - 1},$$

and any strictly increasing function $\rho : \mathbb{R} \to \mathbb{R}$.

Now since $\overline{\dim_{\mathrm{M}}}\mathcal{W} \leq d_{\mathrm{M}}$, for the sequence $\{\delta_k\}_{k \in \mathbb{N}}$, and for a sufficiently large $n$, whenever $k \geq \lfloor\rho(n)\rfloor$, we have

$$\log|N_{\delta_k}(\mathcal{W})| \leq 2d_{\mathrm{M}}\log(\delta_k^{-1})$$
$$= \log(4)d_{\mathrm{M}}k.$$

By splitting the entropy sum in Dudley's tail inequality in two terms, we obtain

$$S_{\mathcal{W}} = \sum_{k \in \mathbb{Z}} \delta_k \sqrt{\log|N_{\delta_k}(\mathcal{W})|}$$
$$= \sum_{k=-\kappa_0}^{\lfloor\rho(n)\rfloor} \delta_k \sqrt{\log|N_{\delta_k}(\mathcal{W})|} + \sum_{k=\lfloor\rho(n)\rfloor}^{\infty} \delta_k \sqrt{\log|N_{\delta_k}(\mathcal{W})|}.$$

For the first term on the right hand side, we use the monotonicity of covering numbers, i.e. $|N_{\delta_k}| \leq |N_{\delta_l}|$ for $k \leq l$, and write

$$\sum_{k=-\kappa_0}^{\lfloor \rho(n) \rfloor} \delta_k \sqrt{\log |N_{\delta_k}(\mathcal{W})|} \leq \sqrt{\log |N_{\delta_{\lfloor \rho(n) \rfloor}}(\mathcal{W})|} \sum_{k=-\kappa_0}^{\lfloor \rho(n) \rfloor} \delta_k$$

$$\leq \sqrt{\log(4)d_{\mathrm{M}} \lfloor \rho(n) \rfloor} \sum_{k=-\kappa_0}^{\infty} \delta_k$$

$$\leq \sqrt{\log(4)d_{\mathrm{M}} \rho(n)} 2^{\kappa_0+1}$$

$$\leq 4\mathrm{diam}(\mathcal{W})\sqrt{\log(4)d_{\mathrm{M}} \rho(n)}.$$

For the second term on the right hand side, we have

$$\sum_{k=\lfloor \rho(n) \rfloor}^{\infty} \delta_k \sqrt{\log |N_{\delta_k}(\mathcal{W})|} \leq \sqrt{\log(4)d_{\mathrm{M}}} \sum_{k=\lfloor \rho(n) \rfloor}^{\infty} \sqrt{k} \delta_k$$

$$\leq \sqrt{\log(4)d_{\mathrm{M}}} \sum_{k=0}^{\infty} k\delta_k$$

$$= 2\sqrt{\log(4)d_{\mathrm{M}}}.$$

Combining these, we obtain

$$S_{\mathcal{W}} \leq 2\sqrt{\log(4)d_{\mathrm{M}}} \left\{ 1 + 2\mathrm{diam}(\mathcal{W})\sqrt{\rho(n)} \right\}.$$

Plugging this bound back in Dudley's tail bound (S39), we obtain

$$\sup_{w,w' \in \mathcal{W}} (\mathcal{G}_n(w) - \mathcal{G}_n(w')) \leq CL\mathrm{diam}(\mathcal{W}) \frac{\sqrt{d_{\mathrm{M}}\rho(n)} + \sqrt{\log(2/\gamma)}}{\sqrt{n}}.$$

Now fix $w_0 \in \mathcal{W}$ and write the triangle inequality,

$$\sup_{w \in \mathcal{W}} \mathcal{G}_n(w) \leq \sup_{w,w' \in \mathcal{W}} (\mathcal{G}_n(w) - \mathcal{G}_n(w')) + \mathcal{G}_n(w_0).$$

Clearly for a fixed $w_0 \in \mathcal{W}$, we can apply Hoeffding's inequality and obtain that, with probability at least $1 - \gamma$,

$$\mathcal{G}_n(w_0) \leq B\sqrt{\frac{\log(2/\gamma)}{n}}.$$

Combining this with the previous result, we have with probability at least $1 - 2\gamma$

$$\sup_{w \in \mathcal{W}} \mathcal{G}_n(w) \leq CL\mathrm{diam}(\mathcal{W}) \frac{\sqrt{d_{\mathrm{M}}}\sqrt{K_{d_{\mathrm{M}}}} + \sqrt{\log(2/\gamma)}}{\sqrt{n}} + B\sqrt{\frac{\log(2/\gamma)}{n}}.$$

Finally replacing and $\gamma$ with $\gamma/2$ and collecting the absolute constants in $c$, we conclude the proof. $\qquad\square$

*Proof of Theorem S2.* The proof follows the same lines of the proof of Theorem 1, except that we invoke Lemma S2 instead of Lemma S1. $\qquad\square$

## Footnotes

[1]We use the multi-index convention $\mathbf{j} = (j_1, \ldots, j_d)$ with each $j_i \in \mathbb{N}_0$, and we use the notation $\partial_w^{\mathbf{j}} \tilde{\Psi}(w, \xi) = \frac{\partial^{j_1} \tilde{\Psi}(w,\xi)}{\partial w_1^{j_1}} \cdots \frac{\partial^{j_d} \tilde{\Psi}(w,\xi)}{\partial w_d^{j_d}}$, and $|\mathbf{j}| = \sum_{i=1}^d j_i$.