[Reviews · NeurIPS 2020]

Review 1

Summary and Contributions: Thank you for the author response. In particular, thanks for clarifying the differences (or lack there of) between the two fractal dimensions (it would be helpful to add such a discussion in the paper). --------------- The paper studies generalization properties of SGD in non-convex problems by modeling its trajectory with a stochastic differential equation (SDE). As their model, the authors use the Feller process, which generalizes a number of previous approaches by allowing for heavy-tailed noise. To bound the generalization error of the method, the authors consider the Haussdorff dimension of the trajectory of the algorithm and they show that generalization scales with this intrinsic dimension as opposed to the actual number of parameters in the model.

Strengths: Overall, the paper is written fairly clearly and I think the main generalization bounds are potentially interesting.

Weaknesses: The two main components of the paper, i.e., the generalization bounds and the Feller process, feel quite disconnected. In particular, the generalization results have little to do with the SDE. The only connection is that the Hausdorff dimension, which appears in the bounds, can be controlled for the Feller process by using results from prior work (Proposition 1, which does not seem very initerpretable). However, it is not at all clear to me that the assumptions introduced in Theorems 1 and 2 are satisfied by the Feller process. Looking at the appendix, the connection turns out to be even weaker, since the generalization bounds are actually shown in terms of the Minkowski dimension as opposed to the Hausdorff dimension, and then Assumption H4 is used to ensure that the two dimensions coincide. Also, for Theorem 2, the additional Assumption H5 is very unintuitive, and it is not clear how restrictive it is.

Correctness: To the best of my knowledge, yes.

Clarity: Yes.

Relation to Prior Work: Yes.

Reproducibility: Yes

Additional Feedback: Line 133 in the appendix says “H1 and 1”. The second “1” should presumably be replaced by something else.


Review 2

Summary and Contributions: After Rebuttal: the authors handle my questions well. I am looking forward to seeing this paper at NeurIPS 2020. ------------------------------------------------------------------------------------------------------------- In this paper, the authors study the generalization property of SGD by modeling its trajectory as feller Process, which includes a broad class of Markov Processes such as Brownian motion and heavy-tailed process. Based on this observation, the authors bring up a novel measurement: Hausdorff dimension, which does not necessarily depend on model capacity, to characterize the complexity of potential trajectories induced by these stochastic learning algorithms including SGD. Generalization error are then bounded using Hausdorff dimension under certain technical assumptions. The authors also perform experiments on deep neural networks. By estimating the Hausdorff dimension qualitatively and comparing it with generalization error, the authors show that Hausdorff dimension indeed play an important role in deciding the generalization performance compared to Euclidean dimension.

Strengths: 1) Good framework. Feller Process is richer enough for us to model the trajectories of various stochastic learning algorithms, including SGD, moment-based algorithms and second-order algorithms. In terms of SGD, it is more realistic compared to Brownian motion: the levy-motion enables large jumps, which might be the reason why SGD can escape bad local minima in practice. 2) Nice complexity measurement. I'm not familiar with chaos theory, but Hausdorff dimension seems to match the fractal structure (generated by SGD) perfectly. On the other hand, Hausdorff dimension, being an intrinsic property of the stochastic process, does not necessarily grow with the model capacity. It is evidently more favorable compared to standard Euclidean dimension.

Weaknesses: 1) Bounded loss seems to be a strong assumption (H1) and crucial in deriving the generalization bound. It is not adequately discussed what additional terms will be involved if this assumption is relaxed. 2) How the learning rate schedule affect the final generalization performance is not adequately discussed in this paper. In terms of good generalization, a recent line of research has demonstrated the importance of initial large learning rate: https://arxiv.org/abs/1907.04595, https://arxiv.org/abs/2003.02218, https://arxiv.org/abs/2002.10365. Is similar phenomenon observed in your experiments?

Correctness: I have checked the derivation of the theory in this paper. The results are sound.

Clarity: Overall, the paper is well-written.

Relation to Prior Work: The relation to prior work has been well-discussed in this paper.

Reproducibility: Yes

Additional Feedback: 1) Please elaborate on the comment "Boundedness of the loss can be relaxed at the expense of using sub-Gaussian concentration bounds and introducing more complexity into the expressions". This issue should be addressed in the Appendix. 2) The authors might want to examine how different learning schedules change the Hausdorff dimension and therefore affect the generalization performance. From a broader perspective, the authors can think about whether or not the theory proposed in this paper is consistent with existing literature in generalization. 3) Personally, I think the authors could add some figures to illustrate the intuition of Hausdorff dimension as well as the fractional behavior of stochastic algorithms such as SGD. The word "roughness" seems vague to me. 4) It seems that the heavy-tailed behavior of SGD is crucial for good generalization. Do you have any theory/hypothesis explaining the underlying mechanism behind such phenomenon?


Review 3

Summary and Contributions: After Rebuttal: The authors did not address my question about the effectiveness of this work in understanding deep neural networks well. And I'd like to keep my socre unchanged, which is positive because of the novelty. ---------------------------------------------------------------------------------- In this paper, the authors provide a generalization bounds for SGD under some assumptions on the training trajectories. They show that the generalization error can be controlled by the Hausdorff dimension of the trajectories if the trajectories can be well-approximated by a Feller process. Their results imply that heavier-tailed process is good for generalization.

Strengths: 1. The authors introduce a novel notation of complexity for the trajectories of a stochastic training algorithm. This is the main contribution of this paper and this metric can be used in the future research. 2.The authors show that the Hausdorff dimension can be used to bound the generalization error bound. This may be useful for us to understanding the success of DNNs.

Weaknesses: I think the experiments in this paper are weak. The authors are recommended to give more results to show that their theories are consistent with the empirical results one observes in practice. For example, it is widely known that DNNs trained with Batch Normalization (BN) can generalize better than those trained without BN. Can the authors use their results to explain it to demonstrate the effectiveness of their results? To be precise, can they characterize the differences between the dynamics of training processes with and without BN and show which of them are much heavier-tailed? In addition, I am not sure whether the assumptions on the training trajectories hold or not in practice. For a theory to understand the success of DNNs, these assumptions need to be verified I think.

Correctness: Yes.

Clarity: Yes.

Relation to Prior Work: Yes.

Reproducibility: Yes

Additional Feedback: Please see my comments on the weaknesses.


Review 4

Summary and Contributions: Update: I'm grateful to the authors for their detailed response to my questions. I have increased my score to reflect this and would be happy to see the paper accepted. ____________________________________________ The authors study an SDE approximation of SGD using heavy-tailed noise and allow for a substantially richer class of processes than previous work. By relating the Hausdorff dimension of these processes to both the heavy-tailedness of the noise and function class capacity, they obtain a bound on generalization that does not depend on the number of parameters.

Strengths: There have been several works that point to the presence of heavy-tailedness in NNs. My previous knowledge of this was that results, especially those connecting to generalization, where empirical. Thus, a clear strength of this work is to provide theoretical results to explain those findings. Moreover, the experimental section of the paper is interesting in that it studies real NN architectures that are able to achieve good performance on standard image classification tasks.

Weaknesses: One question is how much the heavy-tailedness is really buying in the generalization bound in Theorem 1. It seems to yield an improvement of a constant factor over a standard Gaussian SDE bound. If all ones cares about is correlation with generalization perhaps this is enough. The experimental section has some weaknesses also. I think some more extreme, perhaps even toy cases, would be beneficial. For example, some simple cases where the NNs quite clearly radically under- and overfits. While the authors do not highlight it in the main text, their comparison with other capacity metrics in S1 could be improved by considering other datasets and NN architectures, since testing the predictiveness of generalization with different metrics is notoriously tricky.

Correctness: Theorem 1, assumes that \mathcal{Z} is countable. Could the authors discuss this, as it seems false in many common machine learning problems. Assumption H5 seems quite strong and receives little discussion. I understand that it's key to proving the papers results, but I'm unclear whether this assumption is realistic.

Clarity: The language is clear and easy to follow, but overall the structure of the paper could be improved. A lot of time is spent on introduction and background material in the main text. I'm always happy to learn new math, but more emphasis on its importance to the analysis would improve the paper. I think some intuition of how a local assumption of decomposability at a single point, translates into a global result would be helpful.

Relation to Prior Work: I do not know all the literature on SDE models of SGD, but the paper seems well referenced.

Reproducibility: Yes

Additional Feedback: Line 27: "random subset" Random how? Uniformly? Line 228: "Without loss of generality..." Why is this assumption without loss of generality? Line 344: "state-of-the-art" It is not correct to say these models are close to SotA. How was the finite time T chosen in experiments? Do the results say anything about the generalization proerties of NNs trained with full-batch gradient descent?


Review 5

Summary and Contributions: The authors develop a new form of uniform generalization bound over sample paths of continuous-time stochastic optimization models involving Hausdorff dimension. By applying these bounds to models involving heavy-tailed Levy noise (supported by empirical observations in previous work), the paper presents the first theoretical evidence supporting the hypothesis that smaller tail exponents in the distribution of weights implies better generalization, independently of the number of parameters. The precise form of this bound is weakly supported by a few basic numerical experiments.

Strengths: The theoretical results present a significant step forward in validating the connection between generalization and tail exponents of weights in stochastic optimization. To my knowledge, considering the path of the optimizer itself is a new tactic for generalization bounds, and opens a number of exciting possibilities. The model framework considered --- involving Feller processes on R^d --- is highly general (although restricted to continuous time). The presentation is rigorous, and results are detailed. Although basic, the numerical experiments are sufficiently convincing to me that the results have practical relevance.

Weaknesses: Intuition behind the Hausdorff dimension and its importance in the generalization bound (and how it relates to heavy-tailed processes) is sorely lacking. This is especially unfortunate due to the technicality of the work. For example, the authors could mention how the Hausdorff dimension measures clustering/denseness of the sample paths. Or how, unlike light-tailed processes, heavy-tailed processes experience large jumps. Such discussion would help convince readers that the results are not purely academic. The reliance on continuous-time approximations (due to the nature of Hausdorff dimension) is also unfortunate, but could be the subject of future work. Also, the generalization bounds apply only to a finite horizon of the process, ignoring whether the optimization actually converges to a reasonable optimum. This is probably outside the scope of the paper since only the difference between the true risk and empirical risk is of interest, but it still seems relevant given that a strongly heavy-tailed stochastic optimizer may fail to converge to anything.

Correctness: I have encountered no errors in the content of the paper, or its proofs.

Clarity: Due to its technical nature, the paper could be a challenging read for those not already familiar with the subjects discussed. Aside from this and the aforementioned lack of intuition, the paper is quite well-written, and I very much enjoyed reading it.

Relation to Prior Work: The authors have conducted an adequate literature review, and theoretical results pulled from the literature have been appropriately cited and discussed.

Reproducibility: Yes

Additional Feedback: Despite my criticisms, I am very impressed by this paper and believe it provides strong theoretical foundations for future efforts. While the assumptions (particularly H5) are technical and would be too difficult to verify in most cases, I feel this is par for the course in work of this nature. Also, it would be good to highlight an example of a model which is a decomposable Feller process and satisfies the conditions in S3 necessary for Theorem S1 to hold. I'm fairly convinced this is true for quadratic f and constant Sigma, for example, but a comment describing how these assumptions impact the choice of f more generally would be useful. I have a few other comments: Regarding Figure 1, it seems that the slope in the linear trend is different within each depth D. Is it reasonable to compare this slope with some other property of the system suggested by the theory, e.g. a local Lipschitz constant about the optimum? Line 87: `Haussdorf' -> `Hausdorff' Lines 141-153: A very minor point, but I find the `definition' of Feller processes used here a bit odd. Feller processes are processes defined to exhibit the Feller properties --- personally, I find these easier to understand/believe. Unless I'm missing something, it is a subset of these, which have smooth compactly supported functions on R^n as a core, that have a symbol of the form (6) by Courrege's theorem. I understand this is done for convenience, but it is a little confusing for those with outside understanding of Feller processes. Line 289/90: I think this is somewhat disingenuous, as it sounds like the log n term can be improved for free with additional effort. It would be good to mention that this comes at the cost of the log L term. In the supplementary material: Line 86: Why not specify the choice of nu here?

[Author Response · NeurIPS 2020]

We thank all the reviewers for their time and effort to evaluate our paper.

**R1, R3, R4: Assumptions:** We will discuss the assumptions in the context of the SDE in Eq. 4, which is our main
focus. **H1:** The boundedness assumption requires the solutions to the SDE to be 'non-explosive' in the sense that for
every $S$, $\|W_t^{(S)}\| < \infty$ almost surely for all $t \in [0, T]$. As we shortly mentioned in L.269, H1 directly holds under
compact $\mathcal{Z}$ and the conditions of [XZ20, Lem7.1], which requires standard regularity conditions on $f, \Sigma_1, \Sigma_2, \boldsymbol{\alpha}$. **H2.**
only concerns $\ell$. **H3.** follows from Prop.1. and it is not required in Thm2. **H4.** It is easier to analyze the Hausdorff
dimension (HD) of the range of a stochastic process, it has henceforth become the most frequently studied notion of
fractal dimension for Lévy-type processes. Yet, the Minkowski dimension (MD) is more relevant to the generalization
error, hence we used MD as a tool in our proofs and used H4 to ensure it is equal to HD. However, we underline that for
many fractal-like sets, HD is **already** equal to MD (see the discussions in [Mat99, Page 80-81]), which include several
Feller processes like Brownian motion or $\alpha$-stable processes (see [Fal04,Chapter16]). Hence, H4 is not an unrealistic
assumption for our purposes. On the other hand, in Thm.S3, we already proved a bound which does not require H4, but
requires $\mathcal{Z}$ to be finite. **H5:** This assumption is common in statistics (see [Bra83]) but hard to verify in practice. It
essentially quantifies the dependence between $S$ and the trajectory $\{W_t^{(S)}\}_{t \in [0,1]}$, through the constant $M > 0$: small
$M$ indicates that the training error $\hat{\mathcal{R}}$ would not differ much if the training data $S$ is slightly altered. This is similar to
the mutual information used recently in [XR17,arXiv:1806.03803] and to the concept of stability. We agree that H4-5
seem technical, and we will add a more detailed discussion.

**R1: "Prop. 1, ...not...interpretable"** We agree that Prop.1 can be difficult to grasp at a first sight; hence, we provided
a paragraph (L.245) to discuss its semantics in our context, and also we believe that it should be considered as a
contribution as we make this surprising connection with probability theory literature.

**R2: Bounded loss assumption:** In the proof, we require the concentration of empirical risk $\frac{1}{n}\sum_i \ell(w, z_i)$ which, for
a fixed $w$, is a sum of iid r.v.'s $\ell(w, z_i)$. In the current version, we assumed $\ell(w, z_i) \leq B$, and used Hoeffding; however,
the same can be achieved by simply assuming $\exists K, \forall p, \mathbb{E}[\ell(w, z)^p]^{1/p} \leq K\sqrt{p}$, and using sub-Gaussian concentration:
Thms.1-2 still hold with $K$ in place of $B$. **Learning-rate schedule:** We are interested in the relationship between
intrinsic dimensionality and generalization of the deep networks. Relationship between dimensionality and learning rate
is an interesting topic which is out-of-scope for our paper, but a great candidate for future work. **Illustration:** We will
add new figures illustrating the Hausdorff dimension. **Underlying mechanism:** At this stage, we can only speculate
that if the process exhibit heavy-tails near a local minimum, it stays near that minimum even when large jumps occur
due to the heavy-tails (otherwise it would go near a different minimum). In a non-convex setting, this would happen
when the local minimum has a low curvature, connecting our framework to the notion of flatness.

**R3:** Understanding the mechanism of BN is an active research topic, and the relationship between generalization and
BN is not fully understood. Our main focus is the relationship between generalization and intrinsic dimensionality.
Relationship between BN and dimensionality is an interesting future direction which would complement our work.

**R4:** We believe we addressed all the raised issues in what follows. We hope the reviewer could reconsider their overall
score. **Comparison to std. Gaussian SDEs: (1)** The current bounds for Brownian-SDEs (based on isotropic Gaussian
noise) often contain an implicit term, e.g. KL div. (arXiv:1911.02151), which are hard to control and grow with $d$ (see
[NBMS17]), or the sum of step-sizes [MWZZ17] which may result in vacuous bounds (see Fig. 1 in arXiv:1911.02151).
In this sense, our bounds also improve upon existing Brownian-SDE bounds: our result is the first generalization bound
which is independent of $d$ for **both** Brownian and heavy-tailed SDEs. Also our bounds are uniform in the **full path** of
the algorithm; whereas [MWZZ17], arXiv:1911.02151 only control the error at the endpoint, which has to be determined
in advance. **(2)** When the process exhibits heavy-tails, our bounds show that this intrinsic dimension gets even smaller,
can become **arbitrarily** close to zero even when $d$ is large, see Fig1.a. In this sense, we disagree with the reviewer on
the comment "improvement of a constant factor": in our bounds $d_H$ gets very small for networks with large $d$, hence
the improvement is clearly not a constant factor. **Experiments:** We already conducted many experiments on several
neural architectures of different sizes (fully connected, Alexnet, VGG), which all conformed with our theory. However,
we decided to report our results only on VGG for the sake of conciseness, as they contain millions of parameters, which
cannot be explained by existing generalization bounds. Experiments on other architectures produced similar figures. We
will include others in the revised version. On the other hand, The fact that our theory is valid **even** on VGG networks
should be considered as one of our strengths. Moreover, we thank the reviewer for suggesting to experiment the extreme
cases. We agree it would be an interesting addition, we will run the experiments on extreme cases of pure generalization
and pure memorization and add in the final version. **Countability:** Countability of $\mathcal{Z}$ is only assumed in Thm1 (Thm2
does not require it), and it has been considered in various studies, e.g. [BE02]. **Clarity:** We will describe in more detail
why local decomposability reflects in the global behavior. **Add. Comments: (1)** Our theory is agnostic to the way the
minibatch is drawn. **(2)** Contrary to works that exploit the implicit regularization of zero initialization, our bounds are
valid for any fixed initial point, see [arXiv:1707.06618] for similar arguments. **(3)** We will revise and state they are
close to SotA. **(4)** We ran SGD for 100 epochs (see L.340). Note that our bounds do not depend on $T$. **(5)** Thms 1-2
apply to gradient descent as well; however, it is not clear how $\dim_H \mathcal{W}_S$ can be measured in that case.

[Meta-Review · NeurIPS 2020]

The paper studies generalization properties of SGD in non-convex problems by modeling its trajectory with a SDE. To bound the generalization error of the method, the authors consider the Haussdorff dimension of the trajectory of the algorithm. The paper is interesting and technically nontrivial. The paper is timely, as there is a recent interest in heavy-tailed properties of neural network models, and the paper is well written. Author feedback and discussion phase clarified several questions.